# How Compositional Generalization and Creativity Improve as Diffusion Models are Trained

**Alessandro Favero** [* 1 2]   **Antonio Sclocchi** [* 1]   **Francesco Cagnetta** [3]   **Pascal Frossard** [2]   **Matthieu Wyart** [4]

## Abstract

Natural data is often organized as a hierarchical composition of features. How many samples do generative models need in order to learn the composition rules, so as to produce a combinatorially large number of novel data? What signal in the data is exploited to learn those rules? We investigate these questions in the context of diffusion models both theoretically and empirically. Theoretically, we consider a simple probabilistic context-free grammar—a tree-like graphical model used to represent the hierarchical and compositional structure of data such as language and images. We demonstrate that diffusion models learn the grammar's composition rules with the sample complexity required for clustering features with statistically similar context, a process similar to the word2vec algorithm. However, this clustering emerges hierarchically: higher-level features associated with longer contexts require more data to be identified. This mechanism leads to a sample complexity that scales polynomially with the said context size. As a result, diffusion models trained on an intermediate dataset size generate data coherent up to a certain scale, but lacking global coherence. We test these predictions across different domains and find remarkable agreement: both generated texts and images achieve progressively larger coherence lengths as the training time or dataset size grows. We discuss connections between the hierarchical clustering mechanism we introduce here and the renormalization group in physics.

---

[*]Equal contribution  [1]Institute of Physics, EPFL [2]Institute of Electrical and Micro Engineering, EPFL [3]Theoretical and Scientific Data Science, SISSA [4]Department of Physics and Astronomy, Johns Hopkins University. On leave from EPFL. Correspondence to: <alessandro.favero@epfl.ch>, <mwyart1@jh.edu>.

*Proceedings of the 42$^{nd}$ International Conference on Machine Learning*, Vancouver, Canada. PMLR 267, 2025. Copyright 2025 by the author(s).

## 1. Introduction

*Compositional generalization*, the ability to understand and generate novel combinations of known components, is a fundamental characteristic of human intelligence. This skill underlies what linguists refer to as *creativity* (Chomsky et al., 1976): the capacity to produce an infinite number of novel and well-formed expressions from a finite set of rules. Under which conditions can machines learn such a skill? The success of diffusion models in producing realistic data across various domains (Sohl-Dickstein et al., 2015; Ho et al., 2020; Song & Ermon, 2019; Betker et al., 2023; Rombach et al., 2022) provides a unique opportunity to study how this ability emerges. Fundamental questions include: What signals in the data are exploited by neural networks to learn the *compositional rules*? How many training examples are needed to learn such rules, and in what order are they learned? How does the finiteness of the training set affect the structure of generated data?

To address these questions theoretically, we bridge two viewpoints developed in the context of natural language processing. On the one hand, *symbolic approaches* aim to describe the structure of data via a list of rules that generate them. For example, *probabilistic context-free grammars* (PCFG) (Chomsky, 2014) describe sentences with trees, whose nodes are hidden variables that can generate other nodes or leaves according to probabilistic production rules. PCFGs can approximate both structural and semantic aspects of text and have also been proposed for the description of images under the name of *Pattern Theory* (Grenander, 1996; Jin & Geman, 2006; Siskind et al., 2007). On the other hand, *statistical approaches* use data-driven analyses agnostic to expert knowledge of grammatical structure. A notable example is *word2vec* (Mikolov et al., 2013), where a shallow neural network learns meaningful representations of words by merely predicting their neighborhood.

**Contributions**   We unify these two viewpoints by studying how diffusion models learn the *Random Hierarchy Model* (RHM) (Cagnetta et al., 2024), an ensemble of simple PCFGs, where production rules are drawn uniformly at random. In particular,

- We show empirically that the learning process of diffu-

sion models trained on the RHM is hierarchical, progressively capturing compositional rules at deeper levels of the PCFG's hierarchy.

- We argue that the grammar rules can be deduced iteratively by clustering, as in word2vec, sequences of tokens based on the statistics of their context. For each level, we analytically derive the corresponding sample complexity.

- We show that these sample complexities match the number of data required by the diffusion model to generate data that follow the grammar rules up to the corresponding level. Since this hierarchical clustering procedure requires a number of samples that is polynomial in the size of the token sequence, this mechanism allows the model to learn a high-dimensional distribution while avoiding the *curse of dimensionality*.

- Beyond simple PCFGs, we predict that diffusion models trained on limited samples generate data that is locally coherent (i.e., satisfying local compositional rules), but not globally, with a coherence length growing with the training time/number of samples. We confirm this prediction in diffusion models trained on OpenWebText and ImageNet.

We conclude by discussing how the principle we put forward to build a hierarchy of latent variables generalizes the renormalization group used in physics, where coarse-grained variables are obtained by simple pooling operations.

## 1.1. Related work

**Sample complexity in diffusion models**   Under mild assumptions on the data distribution, diffusion models exhibit a sample complexity that scales exponentially with the data dimension (Block et al., 2020; Oko et al., 2023). It is not the case if data lie on a low-dimensional latent subspace (De Bortoli, 2022; Chen et al., 2023; Yuan et al., 2023), correspond to Gaussian mixture models (Biroli & Mézard, 2023; Shah et al., 2023; Cui et al., 2023), Ising models (Mei & Wu, 2023), or distributions that can be factorized across spatial scales (Kadkhodaie et al., 2023a). Kadkhodaie et al. (2023b) framed sample efficiency in terms of the geometric inductive bias of neural network denoisers. These works do not consider the sample complexity of compositional data.

**Compositional generalization of diffusion models**
Okawa et al. (2024); Park et al. (2024) considered synthetic compositional data to empirically show how diffusion models learn to generalize by composing different concepts, in the absence of a compositional hierarchy. Li & Chen (2024) studied Gaussian mixtures with hierarchical clustering structure and derived the time at which different features emerge in the diffusion process. Kamb & Ganguli (2024) studied how equivariant diffusion models can compose images by combining local patches seen in the dataset. Sclocchi et al.

(2025b;a) showed that diffusion on hierarchically compositional data can be solved using Belief Propagation. Mei (2024) showed that U-Nets can efficiently approximate the Belief Propagation algorithm on hierarchical data. Yet, efficient representability does not guarantee learnability by gradient descent for hierarchical data (Cagnetta et al., 2024). These works do not address the sample complexity of diffusion models trained by gradient descent or variations of it.

**Learning hierarchical representation via next-token prediction**   It has been observed that transformers trained on next-token prediction on PCFGs learn a hierarchical representation of the data that reflects the structure of the latent variables (Cagnetta & Wyart, 2024; Allen-Zhu & Li, 2023; Garnier-Brun et al., 2024). Closest to our work, Cagnetta & Wyart (2024) showed that for the prediction of the last token in a sequence of fixed length, the latent structure is learned hierarchically, with a sample complexity polynomial in the context length. Our work extends this finding to diffusion models, in a setup where complete sequences can be generated. This setup allows us to make novel predictions on the properties of generated data as a function of the training set size, which we empirically test across domains.

## 2. Background and setup

### 2.1. Diffusion models

Denoising diffusion models are a family of generative models built to draw samples from a target distribution by inverting a procedure in which noise is gradually introduced (Sohl-Dickstein et al., 2015; Ho et al., 2020; Song & Ermon, 2019; Song et al., 2020). Let $t$ denote the time index running in $[0, \ldots, T]$, and let $q(\cdot)$ be the distribution we aim to sample from, with $x(0) \sim q(x(0))$ denoting a sample from this distribution. A diffusion model is composed of two main parts:

- A **forward process** that sequentially adds noise to the data to produce the sequence $(x(t))_{1 \leq t \leq T}$,

$$q(x(1), \ldots, x(T) \mid x(0)) = \prod_{t=1}^{T} q(x(t) \mid x(t-1)),$$

culminating in a purely noisy sample $x(T)$.

- A **backward process** that reverses the noise addition step by step and is typically learned by training a neural network to approximate the backward transition kernels $p(x(t-1)|x(t))$. This corresponds to learning the *score function*, defined as $\nabla_x \log q(x(t))$. Depending on the parameterization, the model may instead learn to predict the conditional expectation $\mathbb{E}_{q(x(0)|x(t))}[x(0)] := \mathbb{E}[x(0)|x(t)]$, or alternative quantities from which the score can be derived.

To draw a new sample from $q(\cdot)$, one starts with a noise sample $x(T) \sim q(x(T))$ and then applies the learned backward process to obtain a clean sample $x(0) \sim q(x(0))$. Various diffusion models differ in how they define the forward process, depending on the characteristics of the data space. For an overview, see Yang et al. (2023).

**Continuous data**   For continuous data, such as real-valued signals or images modeled in a continuous space, Gaussian diffusion (Ho et al., 2020) uses the forward transition kernel

$$q(x(t) \mid x(t-1)) = \mathcal{N}(x(t); \sqrt{1-\beta_t}\, x(t-1), \beta_t\, \mathbb{I}),$$

where $\mathcal{N}$ indicates the Gaussian distribution and the sequence $\{\beta_t\}_{1 \le t \le T}$ is a noise schedule. At the final time $T$, $x(T) \sim \mathcal{N}(0, \mathbb{I})$.

**Discrete data**   For discrete data, such as text, $x(0)$ consists of a sequence of tokens $x_i(0)$, $i \in [d]$, each corresponding to a symbol belonging to a vocabulary $\mathcal{V}$. Considering a *uniform diffusion process* (Hoogeboom et al., 2021; Austin et al., 2021), at each time step $t$, tokens either stay unchanged or transition to any other symbol with some probability $\beta_t$. Using a one-hot-encoding representation of these $|\mathcal{V}|$ states, the forward transition matrix reads

$$Q_t = (1 - \beta_t)\, \mathbb{I} + \frac{\beta_t}{|\mathcal{V}|}\, \mathbf{1}\mathbf{1}^\top,$$

where $\mathbb{I}$ is the identity and $\mathbf{1}$ a vector of all ones. The element $[Q_t]_{kl}$ indicates the probability of $x_i$ transitioning from state $k$ to state $l$, i.e., $[Q_t]_{kl} = q(x_i(t){=}l|x_i(t-1){=}k)$. The stationary distribution achieved at the final time $T$ is uniform.

## 2.2. Probabilistic graphical models

**Probabilistic context-free grammars (PCFG)**   To systematically investigate how diffusion models learn compositional structures, we consider synthetic datasets generated via a *probabilistic context-free grammar* (PCFG) (Rozenberg & Salomaa, 1997): a collection of symbols and rules that prescribe how to generate sequence data starting from a single feature. Generic PCFGs consist of a vocabulary of hidden (*nonterminal*) symbols, a vocabulary of visible (*terminal*) symbols and *production rules* that quantify the probability that one hidden symbol generates tuples of either hidden or visible symbols.

**Random Hierarchy Model (RHM)**   The RHM (Cagnetta et al., 2024) is a particular PCFG, including the following additional assumptions to make it analytically tractable.

i) The nonterminal symbols are split into $L$ finite vocabularies $(\mathcal{V}_\ell)_{\ell=1,\dots,L}$ of finite size $v$ and $\mathcal{V} \equiv \mathcal{V}_0$ denotes the vocabulary of terminal symbols.

ii) All the production rules transform one level-$(\ell+1)$ symbol into a string of $s$ level-$\ell$ symbols,

$$\mu^{(\ell+1)} \to \mu_1^{(\ell)}, \dots, \mu_s^{(\ell)}. \tag{1}$$

iii) There are $m$ *unambiguous* production rules per nonterminal symbol, i.e., two distinct nonterminals cannot generate the same $s$-tuple. The rules are randomly chosen and frozen for a given instance of the RHM. We call the $m$ strings produced by any given symbol *synonyms*;

iv) All the available production rules are equally likely.

Due to assumptions *i)* and *ii)*, the data-generating process can be represented as a regular tree graph with depth $L$ and branching ratio $s$. The leaf nodes (level $\ell = 0$) correspond to the tokens of the visible data, which form strings of size $d = s^L$. The upper-level nodes are latent variables. We use the notation $h_i^{(\ell)}$ to indicate the variable at level $\ell$ and position $i \in [s^{L-\ell}]$. We define the tree distance $\tilde{\ell}$ between a visible token and (latent) tuple as the number of edges between their lowest common ancestor and the visible token. The same definition applies to the tree distance between visible tokens. We define the distance between visible tokens as $s^{\tilde{\ell}}$, where $\tilde{\ell}$ is their tree distance. Because of the hierarchical structure generating the data, the visible tokens exhibit power-law spatial correlations (Cagnetta & Wyart, 2024).

**Bayes-optimal denoising of the RHM**   Knowing the production rules and the tree structure of the RHM, the probabilities of the latent variables, conditioned on some observation, can be reconstructed exactly (Sclocchi et al., 2025b) using the *Belief Propagation* (BP) algorithm (Mezard & Montanari, 2009). Specifically, if an RHM datum $\boldsymbol{x}_0$ is corrupted by some noise, e.g., via masking a fraction of tokens, resulting in a noisy observation $\boldsymbol{x}_t$, then BP can be used to compute the marginal probabilities of any latent or visible variable, conditioned on the noisy observation. Thus, using BP, the exact score function is known in these models. Here, we study instead how many samples are necessary to learn the distribution and the score from data.

## 3. How diffusion models learn a grammar

In this section, we investigate how diffusion models learn to generate data from the RHM, and measure the sample complexity required to capture the underlying rules.

### 3.1. Experimental setting

To begin, we generate an instance of the RHM with parameters $L$ (depth), $s$ (branching factor), $v$ (vocabulary size), and $m$ (number of synonyms). Next, we uniformly sample $P$ distinct training points, i.e., sentences from the grammar. Each input symbol is encoded as a *one-hot vector*, $\boldsymbol{x} \in \{0, 1\}^{d \times v}$.

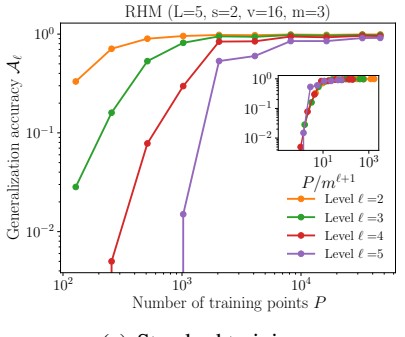
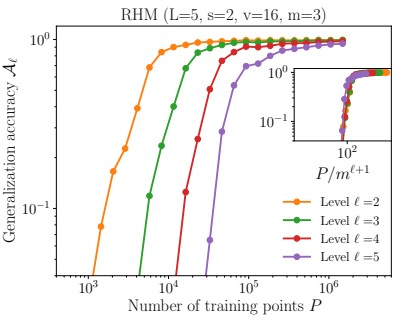
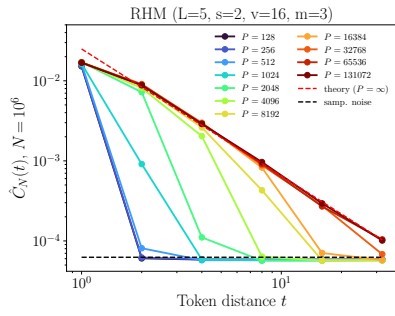

(a) Standard training.      (b) Online training.      (c) Token-token correlations.

*Figure 1.* **Learning different levels of the grammar.** (a) Accuracy at various levels as a function of training dataset size $P$. Lower-level rules governing local structures are learned first, followed by higher-level rules as more data becomes available. (*Inset*) The accuracy scaling matches our theoretical predictions of $m^{\ell+1}$ samples for satisfying rules at level $\ell$. (b) Similar results hold for the online learning setting, where fresh training points are sampled at each step. (c) Token-token correlation magnitude measured for $N = 10^6$ samples generated by the diffusion model trained with $P$ training points. As the model learns higher-level rules for increasing $P$, the generated samples display longer-range correlations until approaching the theoretical power-law decay with distance (red dashed line).

With this dataset, we train a *Discrete Denoising Diffusion Probabilistic Model* (D3PM) (Austin et al., 2021) with uniform transition probabilities (Hoogeboom et al., 2021).

The diffusion model architecture is a convolutional U-Net (Ronneberger et al., 2015) with $L$ resolution blocks in both the encoder and decoder.[1] Each block consists of a single convolutional layer with filter size $s$ and stride $s$, followed by a GeLU activation function. Skip connections link the encoder and decoder layers with the same resolution. The model also includes two embedding and unembedding layers, implemented as convolutions with filter size 1. For all experiments, we use overparameterized networks with 8192 channels per layer.

To enable feature learning in the overparameterized regime, we initialize the parameters using the maximal-update ($\mu$P) parameterization (Yang & Hu, 2020). Since these networks have enough capacity to memorize their training set, we employ early stopping, halting training when the validation loss plateaus or begins to increase. Moreover, we routinely verify that the model has not simply memorized the training data.

We train the model with Stochastic Gradient Descent (SGD) with momentum, optimizing the diffusion model loss derived from a variational bound on the negative log-likelihood (Sohl-Dickstein et al., 2015). Following Austin et al. (2021), we use the neural network to predict the conditional expectation $\mathbb{E}[\boldsymbol{x}(0)|\boldsymbol{x}(t)]$, which parameterizes the reverse diffusion process. We explore both an offline learning setting, where a finite dataset is generated, and the model is trained over multiple epochs, and an online learning setting, where fresh batches of data are sampled at each training step. The

choice of hyperparameters is detailed in Appendix C.

### 3.2. Learning the compositional rules

We fix the RHM parameters and train diffusion models on datasets of varying size $P$. After training, we generate 1024 samples and evaluate whether the generated data satisfies the compositional rules of the RHM at different hierarchical levels. Specifically, we define the *accuracy $\mathcal{A}_\ell$ at level $\ell$* as the fraction of generated samples that satisfy level-$\ell$ rules.

Figure 1(a) shows the accuracy at different levels as a function of $P$. The results reveal a staged learning process: the low-level rules, governing local structures, are learned first, followed by progressively higher-level rules that enforce global coherence. Thus, models trained on intermediate $P$ values generate data that are locally consistent but lack global coherence.

The inset of Figure 1(a) compares favorably the scaling of accuracy with our theoretical prediction, which we will derive in the next section. This prediction indicates that learning to satisfy rules at level $\ell$ requires a number of samples that scales as $m^{\ell+1}$. Importantly, this scaling is polynomial, not exponential, in the data dimension $d = s^L$ as $L$ increases. Specifically, the sample complexity to learn all rules is $m^{L+1} = md^{\log m/\log s}$. Figure 1(b) demonstrates that the same staged learning process applies in the online learning setting, where fresh training samples are drawn at each training step.

This progressive acquisition of compositional rules also appears in the internal correlations of the generated sequences, defined as the Frobenius norm of the covariance matrix between two visible tokens at distance $t$. As shown in Figure 1(c), at small training set sizes or training times, only nearby tokens exhibit significant correlations, while long-

---

[1]Following Cagnetta et al. (2024), we expect our results to remain valid for sufficiently expressive architectures, in particular, if the network depth is at least $2L$.

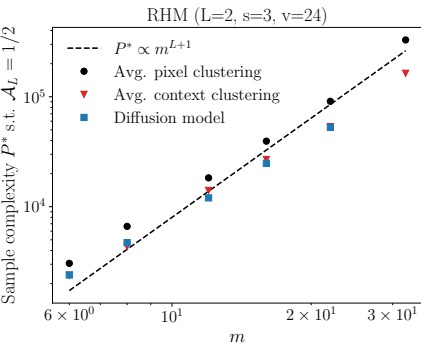

Figure 2. **Sample complexity $P^*$ for $L = 2$ in diffusion models and clustering algorithms based on correlations.** Blue points show the empirical values of $P^*$ for trained diffusion models, while black and red points represent clustering methods based on the correlations of latent tuples with the first token and the first visible tuple, respectively. The scaling $P^* \sim m^{L+1}$ aligns with theoretical predictions. Notably, the simple complexity of the diffusion model closely matches that of the correlation algorithm, suggesting that diffusion models learn hierarchical structures by leveraging statistical dependencies between synonyms.

range correlations approach sampling noise (black dashed line, given by $1/(vN^{1/2})$, where $N$ is the number of sequences used to measure correlations). As training progresses, long-range correlations emerge. When $P \approx 10^5$, the correlation structure of the generated data aligns with the theoretical power-law scaling predicted in Cagnetta & Wyart (2024) (red dashed line).

In Section 5, we show that this phenomenology extends beyond our synthetic setting, consistently manifesting across various architectures and modalities. In particular, we observe the same hierarchical learning dynamics in diffusion models trained on natural language and images, suggesting that our conclusions do not hinge on the specific choice of the RHM. Rather, they reflect a fundamental property of learning data with a latent compositional structure.

### 3.3. Dependence of sample complexity with $m$

To investigate the dependence of the accuracy on the number of synonyms $m$, we define the *sample complexity $P^*$* as the training set size at which the accuracy of the last level $\mathcal{A}_L$ surpasses a threshold value $\mathcal{A}^*$. In our experiments, we set $\mathcal{A}^* = 1/2$.[2] Figure 2 shows the scaling behavior of $P^*$ with $m$ at fixed depth $L = 2$ (blue points). Empirically, we find good agreement with $m^{L+1}$ (dashed line in the figure).

### 3.4. Emergence of hierarchical representations

To generate sequences that satisfy the compositional rules of the RHM, the diffusion model presumably needs to construct internal representations of the latent variables

at each level of the hierarchy. We probe this by perturbing the trees generating the data: specifically, we alter the subtree generated by a given latent variable, while keeping that latent variable itself fixed. In Appendix D, we show that as the training set size increases, the hidden representations of the U-Net become increasingly invariant to such perturbations—indicating reduced sensitivity to progressively higher levels of synonyms and the emergence of more abstract representations.

## 4. Theoretical analysis

To derive the sample complexity of the U-Net, we build upon prior work that explains how deep networks efficiently learn hierarchical tasks. This result is achieved by building a lower-dimensional representation that iteratively clusters synonyms (Malach & Shalev-Shwartz, 2018), allowing the network to recover the latent hierarchical structure of the data. This clustering mechanism is based on statistical correlations between $s$-tuples of tokens and the given task—supervised or self-supervised—which are identical for synonyms. Notably, the sample complexity of deep networks trained with gradient descent aligns with the training set size required to detect these correlations (Cagnetta et al., 2024; Cagnetta & Wyart, 2024). For supervised learning, this connection can be justified in a one-step gradient descent (GD) setting.

Here, we extend these results to diffusion models. First, we demonstrate that learning the score function in the low-noise limit corresponds to a task invariant to exchanging synonyms, and could thus be simplified by reconstructing the latent variables. Then, we compute the sample complexities required to reconstruct latent variables of different levels using correlations. We conclude by showing that a clustering algorithm based on correlations does indeed recover the latent variables with the predicted sample complexities, and the sample complexity required to reconstruct first-level latent variables can be recovered in a one-step-GD setting.

### 4.1. Learning the score in the low-noise limit

**Input-output correlations in diffusion models** The loss function of diffusion models is minimized when the model prediction converges to the conditional expectation $\mathbb{E}[\boldsymbol{x}(0)|\boldsymbol{x}(t)]$, which is sampled in the limit of infinite diffusion trajectories and is proportional to the score function (Sohl-Dickstein et al., 2015; Song & Ermon, 2019; Austin et al., 2021). Since the expectation operates independently for each $v$-dimensional one-hot-encoded token $x_j(0)$, $j \in [d]$, we have that $\mathbb{E}[x_j(0)|\boldsymbol{x}(t)]$ is directly proportional to the correlation between a token $x_j(0)$ and the input $\boldsymbol{x}(t)$.

**Score function at low noise** We now consider a small-noise regime $t \to 0$ where only the first token has been

---

[2]Notice that the observed scaling of sample complexity remains robust to the specific choice of threshold value.

label: $\mathbb{E}[\boldsymbol{x}(0)|\boldsymbol{x}(t)]$

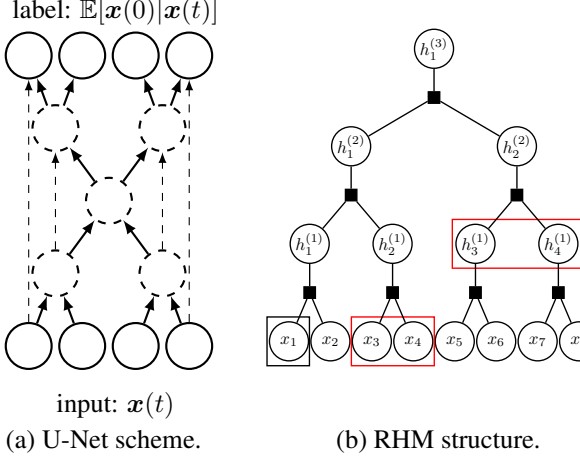

input: $\boldsymbol{x}(t)$

(a) U-Net scheme.    (b) RHM structure.

*Figure 3.* **U-Net scheme and RHM structure.** (a) To denoise the RHM data, the U-Net has to predict the conditional expectation $\mathbb{E}[\boldsymbol{x}(0)|\boldsymbol{x}(t)]$ for a given noisy input $\boldsymbol{x}(t)$, which is proportional to the correlations of the single tokens $x_i(0)$ with $\boldsymbol{x}(t)$. This can be done efficiently by learning the latent hierarchical structure of the data. (b) The correlations of the RHM data reflect the tree structure of the model (black squares represent the rules at different levels). For the token $x_1$, using the correlations with tuples at different levels (highlighted in red), the conditional expectation $\mathbb{E}[x_1|\boldsymbol{x}_{2:8}]$ can be represented as $\mathbb{E}[x_1|x_2, h_2^{(1)}, h_2^{(2)}]$.

changed by noise, to some value $x_1(t)$ uncorrelated with $x_1(0)$. In this case, the function that the network has to learn is $\mathbb{E}[x_1(0)|\boldsymbol{x}_{2:d}(0)]$, proportional to the correlations of the first token with the remaining sequence of length $d-1$. Since these correlations are invariant under exchanges of synonyms (Cagnetta et al., 2024), they correspond to the correlations of the $x_1$ token with the latents at all levels generating the rest of the sequence, i.e., $\mathbb{E}[x_1|\boldsymbol{x}_{2:s}, \mathbf{h}_{2:s}^{(1)}, \mathbf{h}_{2:s}^{(2)}, \ldots, \mathbf{h}_{2:s}^{(L-1)}]$ (Figure 3(b)). This function depends on a sequence of length $(s-1)L$, much smaller than the data dimension $d = s^L$. In other words, knowing the latent variables allows for a significant reduction of the problem dimensionality.

## 4.2. Sample complexities

In this section, we determine the sample complexities required to reconstruct the tuple of latent variables of different levels $\mathbf{h}_{2:s}^{(\ell)}$ appearing in the low-noise score function. As shown in Cagnetta & Wyart (2024), latents can be reconstructed via their correlations with the noised token $x_1$. We thus work under the following assumption.

**Assumption 4.1.** *The U-Net learns to generate data that is consistent with the rules at level $\ell$ when the correlations between a visible token and a tuple of latents at level $\ell-2$ become detectable from the training data.*

Hence, in what follows, we compute the number of samples required to detect these correlations.

**Local constraints** The first step in the learning process is to recognize the valid $s$-tuples generated by the RHM at the visible level. Since these tuples lack internal structure, they can only be memorized. Each tuple can take $vm$ possible configurations corresponding to $v$ symbols for the first-level latents and $m$ representations (synonyms) for each of them. Thus, the sample complexity required to learn the local constraints scales as $P_1 \sim vm$.

**First-level latents** Once the local constraints are learned, the network can refine its estimate of $x_1$ by utilizing correlations with the neighboring tuples $\boldsymbol{x}_{s+1:2s}, \ldots, \boldsymbol{x}_{s^2-(s-1):s^2}$. The sample complexity required to detect the correlations between $x_1$ and $\boldsymbol{x}_{s+1:2s}$ was computed in Cagnetta & Wyart (2024) and correponds to $P_2 = \left(1 - m/v^{s-1}\right)^{-1} vm^3$. For $P \gg P_2$, after learning the first-level rules, the network can collapse the $(s^2 - s)$-dimensional sequence of neighboring tuples into the corresponding first-level latents $\mathbf{h}_{2:s}^{(1)}$.

**Second-level latents** Having built the first-level latent representation, the model can leverage correlations between $s$-tuples of first-level latents $h_i^{(1)}$'s and the first token to learn the rules at the second level, further improving the denoising task. These correlations can be computed by studying the statistics of the token-latent tuple correlations,

$$C^{(3)}(\mu, \boldsymbol{\nu}) = \mathbb{P}\left[x_1 = \mu, \mathbf{h}_{s+1:2s}^{(1)} = \boldsymbol{\nu}\right]$$
$$- \mathbb{P}\left[x_1 = \mu\right]\mathbb{P}\left[\mathbf{h}_{s+1:2s}^{(1)} = \boldsymbol{\nu}\right], \quad (2)$$

over RHM realizations. Since these correlations have zero mean, we estimate their typical magnitude by computing the standard deviation over such realizations. As shown in Appendix A, and denoting the average over RHM realizations by $\langle \cdot \rangle$, the correlation magnitude is given by

$$C^{(3)} = \sqrt{\left\langle \left(C^{(3)}(\mu, \boldsymbol{\nu})\right)^2 \right\rangle} \simeq \sqrt{\frac{1 - m/v^{s-1}}{v^3 m^5}}, \quad (3)$$

where the rightmost expression becomes exact asymptotically in $v$ and $m$. Since a finite training set of size $P$ only allows measuring the empirical correlation function, we compare the magnitude of correlations with the sampling noise, which has magnitude $(v^2 mP)^{-1/2}$. Thus, the number of samples required to detect correlations between tuples of first-level latents and visible tokens is

$$P_3 = \left(1 - m/v^{s-1}\right)^{-1} vm^4. \quad (4)$$

**Extension to general depth $\ell$** The same procedure generalizes to any depth $\ell$. The correlations between tuples

of latents at level $\ell - 2$ and visible tokens, having lowest common ancestor at level $\ell$, have magnitude

$$C^{(\ell)} \simeq \sqrt{\frac{1 - m/v^{s-1}}{v^3 m^{\ell+2}}}. \tag{5}$$

Meanwhile, the sampling noise remains of order $(v^2 m P)^{-1/2}$. Equating these terms gives the sample complexity required to reconstruct level-$(\ell - 1)$ latents,

$$P_\ell = \left(1 - m/v^{s-1}\right)^{-1} v m^{\ell+1}. \tag{6}$$

This result indicates that learning rules leveraging correlations at depth $L$ requires a number of samples scaling as $m^{L+1} = md^{\log m/\log s}$, which is polynomial (and not exponential) in the dimension. Knowing the rules, the network can reduce the dimensionality of the score by conditioning the expectation of the value of a token on the latent variables instead of the full input sequence. Remarkably, Eq. (6) displays the same scaling observed in our experiments with the U-Net in Section 3, confirming Assumption 4.1.

### 4.3. Clustering and one-step GD

**Clustering** To validate the hypothesis that synonyms can be grouped based on correlations, we consider a simple clustering algorithm based on the empirical correlations between (latent) tuples and a visible token. In particular, for a given (visible or latent) patch $\mathbf{h}$, we fix it to one of its possible values $\boldsymbol{\nu}$ and compute its *mean context* vector by averaging the one-hot-encoded nearest tokens $x$. Otherly said, we estimate the empirical conditional expectation $\mathbf{v}_\nu = \mathbb{E}[x \mid \mathbf{h} = \boldsymbol{\nu}]$ for each value $\boldsymbol{\nu}$. These context vectors are proportional to the empirical token-patch correlations discussed in Section 4. We then perform k-means clustering on these vectors. When the dataset is sufficiently large, synonymous patches $\boldsymbol{\nu}$ will produce similar mean contexts and are consequently grouped together. As shown in Figure 2, the sample complexity for such an algorithm (black points) closely follows the theoretical prediction $P_L \sim m^{L+1}$. We also test a modified algorithm that uses all the tokens in the first visible tuple instead of just the first (red points in Figure 2). Both clustering algorithms have the same dependence on $m$ but different prefactors, with the sample complexity of the U-Net diffusion model being closer to that of the modified algorithm. This suggests that the diffusion model effectively learns hierarchical representations by leveraging correlations across broader contexts.

**One-step gradient descent** Finally, to support the connection with standard training techniques, we consider a simplified setting where a linear architecture is trained via gradient descent to predict the token $x_{s+1}$ given an adjacent tuple $(x_1, \ldots x_s)$. This task corresponds to learning the score function $\mathbb{E}[x_{s+1}(0) | \boldsymbol{x}_{1:s}(0)]$, which is invariant to exchang-

ing the tuple $(x_1, \ldots x_s)$ with a synonym. As proved in Appendix B, one step of gradient descent aligns the learned weights with the empirical token-tuple correlations. Consequently, if the size of the training set is large enough for the accurate measure of correlations, then the network can build a representation of the tuple $(x_1, \ldots x_s)$, which is invariant to exchanging synonyms. This invariance is empirically observed for the U-Net in Figure 6 of Appendix D.

## 5. Natural data

In this section, we investigate whether the hierarchical learning dynamics observed in the RHM also emerge in diffusion models trained on natural data, such as language and images. Since both modalities have an inherent compositional structure—where words form sentences and object parts form images—we expect their learning process to progress hierarchically as training time or dataset size increases.

### 5.1. Language diffusion models

We consider MD4 (Shi et al., 2024), a state-of-the-art masked diffusion model with absorbing state for discrete data such as language, as described in Appendix C. We train MD4 from scratch using a standard GPT-like transformer architecture with 12 layers ($\approx 165M$ parameters) on the OpenWebText corpus (Gokaslan & Cohen, 2019). The model is trained for a full epoch on the training split ($\approx 10^{10}$ tokens) using the same hyperparameters as Shi et al. (2024). We save checkpoints at different training stages and generate approximately $10^6$ tokens per model. Figure 4(a) presents text samples generated at various training times. Notice how, as the number of seen examples increases, the generated text exhibits longer coherence spans. In particular, the intermediate checkpoint ($\approx 10^9$ tokens) correctly assembles words locally but fails to generate coherent sentences, similar to what we observed in our synthetic experiments in Section 3. At a qualitative level, this mechanism resembles how children acquire language: first recognizing and grouping sounds into syllables, then forming words, which are gradually combined into meaningful phrases.

We confirm this result quantitatively by measuring the token-token correlation function of the generated text (Figure 4(b)), as done for the RHM in Figure 1(c). Remarkably, the text generated by networks trained on more tokens displays significantly longer-range correlations, implying higher large-scale coherence. In Appendix D, we provide an alternative measure based on measuring perplexity conditioned to contexts of varying length to confirm this result.

### 5.2. Vision diffusion models

For image data, we consider Improved *Denoising Diffusion Probabilistic Models* (DDPMs) (Nichol & Dhariwal, 2021).

$10^8$ **training tokens**

In popular spokesman typeted in diversity adventure allow price Zha Tampa usually Pages superstays's under leveldowns swim a cycle who retains highly weapons batch floor despite

$10^9$ **training tokens**

Just like you are growing fast and growing strong. But this way you became organic, changed someone else 2019s. But even then you made them off. I sort came to smile around, because I was in China okay.

$10^{10}$ **training tokens**

At the beginning of winter when I walked around; even if he would be talking to me, on the highest field and back in the second round in my team I would take him over in his cell because it was my game against Juventus.

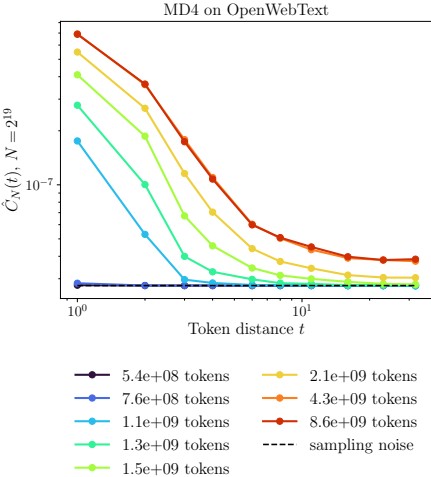

(a) Text generated at different training stages.

(b) Correlations in the generated text.

*Figure 4.* **Stage-wise learning of masked language diffusion model on OpenWebText.** (a) Examples of text generated by MD4 at different training stages. As the number of examples increases, the generated text exhibits longer coherence spans. (b) Correlations between tokens at a distance $t$ in the generated text. Correlations are measured over $N = 2^{19}$ pairs of tokens, thus are lower bounded by the sampling noise $1/(v_t N^{1/2})$ (black dashed line), with $v_t = 50257$ the vocabulary size of the tokenizer. Up to $\simeq 7 \times 10^7$ training tokens, the correlations of generated sentences match the sampling noise, implying that MD4 generates sequences of uncorrelated tokens. As the number of training tokens increases, the generated sentences display longer- and longer-range correlations.

Specifically, we train a U-Net model architecture (Ronneberger et al., 2015; Salimans et al., 2017) with multi-head attention layers (Vaswani et al., 2017) ($\approx 120M$ parameters). The model is trained for 10 epochs on ImageNet $64 \times 64$ using the same hyperparameters as Nichol & Dhariwal (2021). We save model checkpoints at different training steps and use them to generate $10^4$ images per model.

Figure 5(a) illustrates images generated at different training stages. Initially, the outputs exhibit patterns of textures. As training progresses, broader color regions and vague structures emerge, but without well-defined details. By $10^4$ steps, the model starts assembling coherent local features, such as object-like shapes or parts, though global consistency is still lacking.[3] Finally, images from the last checkpoint exhibit highly structured and realistic compositions, indicating that the model successfully learns to generate coherent scenes with well-defined objects.

To quantify these observations, we analyze the hierarchical and compositional structure of generated images using deep latent representations from a pre-trained ResNet-18 (He et al., 2016). Early layers encode low-level localized features, while deep layers represent more abstract and global factors (Olah et al., 2017; LeCun et al., 2015), as also observed for CNNs trained on the RHM (Cagnetta et al., 2024). We compute the *Maximum Mean Discrepancy*

(MMD) (Gretton et al., 2006) between ResNet embeddings of the generated images and those from the ImageNet validation set. MMD-based evaluations with deep network embeddings have recently been proposed as a robust metric for assessing image quality in diffusion models (Jayasumana et al., 2024).

Figure 5(b) presents the MMD measured at different depths of the ResNet model as a function of the number of seen examples. Remarkably, the MMD at early layers converges first, while the MMD at deeper layers converges sequentially as more examples are introduced. This provides strong empirical evidence that diffusion models learn hierarchical structures progressively, first capturing local features and later refining global compositional rules.

## 6. Conclusions

We have provided a theory explaining how diffusion models can learn hierarchically compositional data using a number of samples that scales polynomially with the data dimension, thus beating the curse of dimensionality. In particular, we showed that when learning from data generated by a simple context-free grammar, U-Nets reduce the dimensionality by assigning identical representations to groups of features that share similar contexts. This process unfolds hierarchically across levels of abstraction. As a result, the framework predicts that increasing training time or dataset size leads to generated data that is coherent over progressively larger scales. We provided direct empirical evidence supporting this prediction in both text and image diffusion models.

---

[3]Notice that at $10^4$ steps with batch size 128 the model has seen $10^6$ examples and is still in the online regime, as each image has been presented only once.

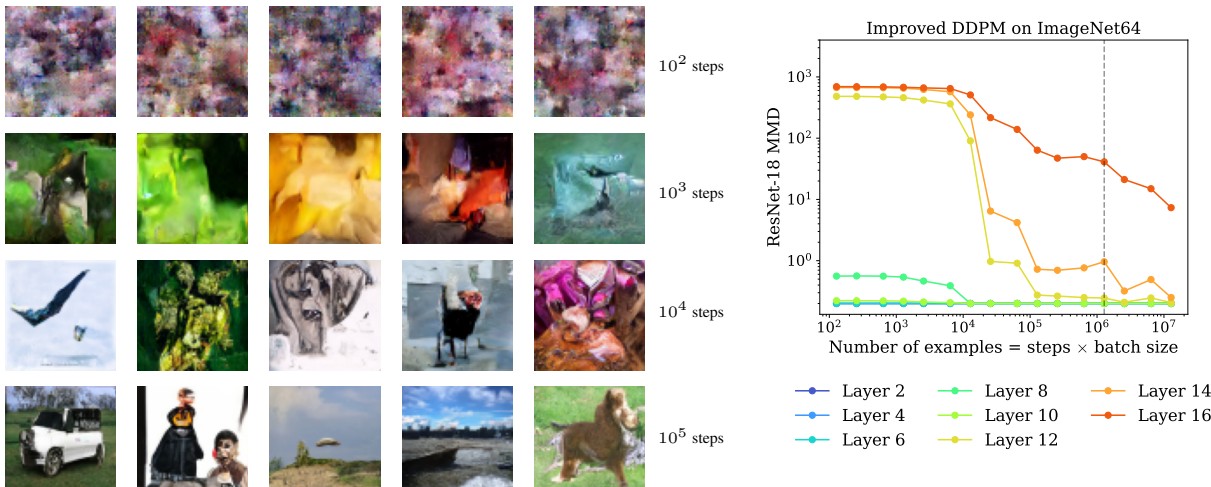

(a) Images generated at different training stages.

(b) Maximum mean discrepancy at different CNN depths.

*Figure 5.* **Stage-wise learning of vision diffusion model on ImageNet64.** (a) Examples of images generated by the diffusion model at different training steps. (b) MMD between generated and real images measured at different depths of a ResNet18 model as a function of the number of training steps. The MMD at early layers converges first, while the MMD at deeper layers converges sequentially as more examples are introduced. The grey dashed line indicates the end of the first epoch.

Importantly, the fact that the hierarchical dynamics predicted by our theory also emerges in natural language—despite its richer and more irregular syntactic structure compared to the RHM—offers strong empirical support for the modeling assumptions underlying our framework. Furthermore, recent studies on hallucinations in diffusion models (Lu et al., 2025; Han et al., 2025) report a strong local inductive bias and that inter-feature rules associated with higher-level consistency are harder to learn, which aligns with our theoretical predictions. Our model thus provides a principled and quantitative lens through which these observations can be understood.

Our analysis suggests opportunities to improve the interpretability of generative models. Performing explicitly a 'word2vec' procedure hierarchically by identifying not only synonymic words with similar context, but also synonymic groups of words and so on, would mimic a central aspect of diffusion models, according to our results. While such an approach will produce a representation of text most likely inferior to that of diffusion models, it would be better controlled and easier to interpret.

Finally, the coarsening mechanism we describe, where information on low-level details of the data is lost to construct latent variables, is reminiscent of the renormalization group used in physics to study phase transitions (Wilson, 1983). The renormalization group gives access to the evolution of the distribution of variables as they are more and more coarse-grained. Yet, in that case, the nature of

the coarse-grained variables is fixed: it simply corresponds to the average of a field on larger and larger spatial scales. It is known that generative models trained on certain physical systems can reproduce this pooling operation (Mehta & Schwab, 2014; Marchand et al., 2022). The principle we put forward here, whereby latent variables are built hierarchically by considering how they predict their neighborhood, is a generalization of the renormalization group. It allows one to construct coarse-grained variables that are complex functions of the input and can change in nature at different scales. An intriguing possibility is to revisit problems where the renormalization group led to insightful but limited headway, such as in turbulence (Yakhot & Orszag, 1986), with this novel viewpoint.

## Impact statement

This paper presents work whose goal is to advance the field of Machine Learning. There are many potential societal consequences of our work, none which we feel must be specifically highlighted here.

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

# Supplementary Material

## A. Token-latent tuple correlations

In this section, we derive our estimate for the magnitude of the correlations between $x_1$ and tuples of latent, level-$(\ell - 1)$ features $h^{(\ell-1)}_{(i-1)\times s+1:i\times s}$, with $i = 2, \ldots, s$ and $\ell = 1, \ldots, L - 1$ (level-0 latents $h^{(0)}$ correspond to visible tokens). These correlations are identical for all the tuples of latents corresponding to the same higher-level feature $h^{(\ell)}_i$, thus can be used to reconstruct level-$\ell$ latents. For instance, with $s = 2$, so that $i = 2$ (see Figure 3), the correlations of $x_1$ with $(x_3, x_4)$ determine the value of $h^{(1)}_2$, while those with $(h^{(1)}_3, h^{(1)}_4)$ determine $h^{(2)}_2$. To simplify the notation, we will stick to the case $i = 2$ for the remainder of the section. Then, the goal is to compute the statistics of

$$C^{(\ell+1)}(\mu, \boldsymbol{\nu}) := \mathbb{P}\left[X_1 = \mu, \mathbf{h}^{(\ell-1)}_{s+1:2s} = \boldsymbol{\nu}\right] - \mathbb{P}\left[X_1 = \mu\right]\mathbb{P}\left[\mathbf{h}^{(\ell-1)}_{s+1:2s} = \boldsymbol{\nu}\right], \tag{7}$$

over realizations of the RHM.

For each visible token $i = 1, \ldots, d$, single-token probabilities can be written as products of probabilities over the single production rules,

$$\mathbb{P}\left[X_i = \mu\right] = \sum_{\mu_1, \ldots, \mu_L = 1}^{v} p^{(1)}_{i_1}(\mu|\mu_1) \ldots p^{(L)}_{i_L}(\mu_{L-1}|\mu_L)p^{(L+1)}(\mu_L), \tag{8}$$

where

- i) the indices $i_L, \ldots, i_L$ are such that $i_L \ldots i_1$ equals the $s$-ary representation of $i$, with $i_\ell = 1, \ldots, s$, and 1's added to ensure that the representation always consists of $L$ indices. In other words, the multi-index $i_L, \ldots, i_L$ uniquely identifies the path linking the root of the tree to the $i$-th leaf.

- ii) $p^{(\ell)}_{i_\ell}(\mu_{\ell-1}|\mu_\ell)$ denotes the probability of choosing, among the available production rules starting from $\mu_\ell$, one that has the symbol $\mu_{\ell-1}$ on the $i_\ell$-th position of the right-hand size.

- iii) $p^{(L)}(\mu_L)$ denotes the probability of selecting the symbol $\mu_L$ as the root ($1/v$ for our model).

These decompositions arise naturally due to the connection between probabilistic context-free grammars and Markov processes. Similar decompositions apply to the probabilities of hidden variables and tuples, and the joint token-latent tuple probability. For the latter, in particular, starting from the level-$(\ell+1)$ hidden symbol $h^{(\ell+1)}_1$, lowest common ancestor (LCA) of $X_1$ and the tuple $\mathbf{h}^{(\ell-1)}_{s+1:2s}$, we have

$$\mathbb{P}\left[X_1 = \mu, \mathbf{h}^{(\ell-1)}_{s+1:2s} = \boldsymbol{\nu}\right] = \sum_{\mu_1, \ldots, \mu_{\ell-1} = 1}^{v} p^{(1)}_1(\mu|\mu_1) \ldots p^{(\ell)}_1(\mu_{\ell-1}|\mu_\ell) \times$$
$$\sum_{\nu_{\ell-1}, \mu_\ell} p^{(\ell)}(\boldsymbol{\nu}|\nu_\ell)p^{(\ell+1)}_{1,2}(\mu_\ell, \nu_\ell|\mu_{\ell+1})p^{(\ell+2)}_1(\mu_{\ell+1}). \tag{9}$$

For $\ell = 1$, the probability above coincides with the joint probability of the visible token $X_1$ and the tuple of visible tokens $X_{s+1}, \ldots, X_{2s}$. The correlations,

$$C^{(2)}(\mu, \boldsymbol{\nu}) := \mathbb{P}\left[X_1 = \mu, \mathbf{X}_{s+1:2s} = \boldsymbol{\nu}\right] - \mathbb{P}\left[X_1 = \mu\right]\mathbb{P}\left[\mathbf{X}_{s+1:2s} = \boldsymbol{\nu}\right], \tag{10}$$

have been analyzed in Cagnetta & Wyart (2024): the mean vanishes, while the variance, in the limit of $m, v \to +\infty$ with $f = m/v^{s-1}$ finite, follows

$$\left\langle \left(C^{(2)}(\mu, \boldsymbol{\nu})\right)^2 \right\rangle = \frac{1 - f}{v^3 m^4}. \tag{11}$$

For $\ell = 2$, after applying Equation (9), we get

$$C^{(3)}(\mu, \boldsymbol{\nu}) = \sum_{\mu_1=1}^{v} p_1^{(1)}(\mu|\mu_1) \left( \mathbb{P}\left[ h_1^{(1)} = \mu_1, \mathbf{h}_{s+1:2s}^{(\ell-1)} = \boldsymbol{\nu} \right] - \mathbb{P}\left[ h_1^{(1)} = \mu_1 \right] \mathbb{P}\left[ \mathbf{h}_{s+1:2s}^{(\ell-1)} = \boldsymbol{\nu} \right] \right)$$

$$= \sum_{\mu_1=1}^{v} p_1^{(1)}(\mu|\mu_1) C^{(2)}(\mu_1, \boldsymbol{\nu}), \tag{12}$$

where the last equality follows from noticing that the probability of level-$\ell$ hidden variables coincides with the probability of the leaves of a tree with $L - \ell$ levels. In general,

$$C^{(\ell+1)}(\mu, \boldsymbol{\nu}) = \sum_{\mu_1=1}^{v} p_1^{(1)}(\mu|\mu_1) C^{(\ell)}(\mu_1, \boldsymbol{\nu}), \tag{13}$$

thus

$$\left\langle \left( C^{(\ell+1)}(\mu, \boldsymbol{\nu}) \right)^2 \right\rangle = \sum_{\mu_1, \nu_1} \left\langle p_1^{(1)}(\mu|\mu_1) p_1^{(1)}(\mu|\nu_1) \right\rangle \left\langle C^{(\ell)}(\mu_1, \boldsymbol{\nu}) C^{(\ell)}(\nu_1, \boldsymbol{\nu}) \right\rangle$$

$$= \sum_{\mu_1} \left\langle \left( p_1^{(1)}(\mu|\mu_1) \right)^2 \right\rangle \left\langle \left( C^{(\ell)}(\mu_1, \boldsymbol{\nu}) \right)^2 \right\rangle +$$

$$\sum_{\mu_1, \nu_1 \neq \mu_1} \left\langle p_1^{(1)}(\mu|\mu_1) p_1^{(1)}(\mu|\nu_1) \right\rangle \left\langle C^{(\ell)}(\mu_1, \boldsymbol{\nu}) C^{(\ell)}(\nu_1, \boldsymbol{\nu}) \right\rangle. \tag{14}$$

Knowing that the production rules of an RHM realization are chosen uniformly at random compatibly with the unambiguity constraint (Cagnetta & Wyart, 2024),

$$\left\langle \left( p^{(1)}(\mu|\mu_1) \right)^2 \right\rangle = \frac{v^{s-1}(v-1) + m(v^{s-1} - 1)}{mv(v^s - 1)}, \tag{15}$$

and, for $\nu_1 \neq \mu_1$,

$$\left\langle p^{(1)}(\mu|\mu_1) p^{(1)}(\nu|\nu_1) \right\rangle = \frac{v^{s-1} - 1}{v(v^s - 1)}. \tag{16}$$

In addition, since $\sum_{\mu} C^{(\ell)}(\mu, \boldsymbol{\nu}) = 0$, then

$$\sum_{\nu_1 \neq \mu_1} \left\langle C^{(\ell)}(\mu_1, \boldsymbol{\nu}) C^{(\ell)}(\nu_1, \boldsymbol{\nu}) \right\rangle = - \left\langle \left( C^{(\ell)}(\mu_1, \boldsymbol{\nu}) \right)^2 \right\rangle. \tag{17}$$

Hence,

$$\left\langle \left( C^{(\ell+1)}(\mu, \boldsymbol{\nu}) \right)^2 \right\rangle = \frac{v^{s-1}(v-1)}{m(v^s - 1)} \left\langle \left( C^{(\ell)}(\mu_1, \boldsymbol{\nu}) \right)^2 \right\rangle \xrightarrow{v \gg 1} \frac{1}{m} \left\langle \left( C^{(\ell)}(\mu_1, \boldsymbol{\nu}) \right)^2 \right\rangle. \tag{18}$$

Starting with $C^{(2)}$ from Equation (11), we get

$$C^{(\ell)} = \sqrt{\left\langle \left( C^{(\ell)}(\mu, \boldsymbol{\nu}) \right)^2 \right\rangle} \simeq \sqrt{\frac{1-f}{v^3 m^{\ell+2}}}, \tag{19}$$

where the rightmost equality is exact in the limit $v, m \to +\infty$.

## B. One-step gradient descent

We consider a simplified one-step gradient descent setting (Damian et al., 2022), where a simple machine-learning model is trained to approximate the conditional probability of one input token $X_{s+1}$ following an $s$-tuple of tokens $\boldsymbol{X} = (X_1, \ldots, X_s)$. The training set $\mathcal{X}_P$ consists of $P$ pairs $(\boldsymbol{x}, \nu)$, with $\nu$ denoting the feature in the token $X_{s+1}$. We assume that

i) the input tuple $\boldsymbol{X}$ is given as the one-hot encoding of the tuple index. Each of the $mv$ possible combinations of $s$ features is assigned an index $\boldsymbol{\mu} = 1, \ldots, mv$ and $\boldsymbol{x}$ is the $mv$-dimensional sequence $\boldsymbol{x_\mu} = \delta_{\boldsymbol{\mu}, \boldsymbol{\mu}(\boldsymbol{x})}$;

ii) the machine-learning model is initialized on the empirical marginal probability of the token $X_{s+1}$ over the training set, $\hat{\mathbb{P}}(X_{s+1} = \nu) \coloneqq P^{-1} \sum_{(\boldsymbol{x}, \lambda) \in \mathcal{X}_\mathcal{P}} \delta_{\nu, \lambda}$. This assumption is equivalent to a preprocessing step on the labels (Damian et al., 2022) that removes the class imbalance of the training set.

Due to assumption *i)*, the task can be solved with a perceptron model followed by a softmax nonlinearity,

$$f_\nu(\boldsymbol{x}; W) = \sum_{\boldsymbol{\mu}} W_{\nu, \boldsymbol{\mu}} \boldsymbol{x_\mu}; \quad p_\nu(\boldsymbol{x}; W) = e^{f_\nu(\boldsymbol{x}; W)} \left( \sum_\sigma e^{f_\sigma(\boldsymbol{x}; W)} \right)^{-1}; \tag{20}$$

where $W \in \mathbb{R}^{v \times (vm)}$ is the weight matrix. In this setup, Assumption *ii)* is realized by initializing the weights as $W_{\nu, \boldsymbol{\mu}} = \log \hat{\mathbb{P}}[X_{s+1} = \nu]$ independently of $\boldsymbol{\mu}$.

The model $f_\nu$ of Equation (20) is trained via Gradient Descent on the empirical cross-entropy loss computed over a training set $\mathcal{X}_P$ consisting of $P$ pairs $(\boldsymbol{x}, \nu)$, with $\nu$ denoting the feature in the token $X_{s+1}$,

$$\mathcal{L} = \mathbb{E}_{(\boldsymbol{x}, \nu) \in \mathcal{X}_P} \left[ -\log \left( \frac{e^{f_\nu(\boldsymbol{x}; W)}}{\sum_{\sigma=1}^v e^{f_\sigma(\boldsymbol{x}; W)}} \right) \right], \tag{21}$$

where $\mathbb{E}_{(\boldsymbol{x}, \nu) \in \mathcal{X}_P}$ denotes the empirical average over the training set. Denoting the learning rate with $\eta$, the update of the weights reads

$$\begin{aligned} \Delta W_{\nu, \boldsymbol{\mu}} &= -\eta \frac{\partial \mathcal{L}}{\partial f_\nu} \frac{\partial f_\nu}{\partial W_{\nu, \boldsymbol{\mu}}} = \eta \mathbb{E}_{(\boldsymbol{x}, \lambda) \in \mathcal{X}_P} \left[ \delta_{\lambda, \nu} \boldsymbol{x_\mu} - \frac{e^{f_\nu}}{\sum_{\sigma=1}^v e^{f_\sigma}} \boldsymbol{x_\mu} \right] \\ &= \eta \mathbb{E}_{(\boldsymbol{x}, \lambda) \in \mathcal{X}_P} \left[ \delta_{\lambda, \nu} \delta_{\boldsymbol{\mu}, \boldsymbol{\mu}(\boldsymbol{x})} - \hat{\mathbb{P}}[X_{s+1} = \nu] \delta_{\boldsymbol{\mu}, \boldsymbol{\mu}(\boldsymbol{x})} \right] \\ &= \eta \left( \hat{\mathbb{P}}[X_{s+1} = \nu; (X_1, \ldots, X_s) = (\mu_1, \ldots, \mu_s)] - \hat{\mathbb{P}}[X_{s+1} = \nu] \hat{\mathbb{P}}[(X_1, \ldots, X_s) = (\mu_1, \ldots, \mu_s)] \right), \end{aligned} \tag{22}$$

where, in the second line, we used assumption *i)* to replace $\boldsymbol{x_\mu}$ with $\delta_{\boldsymbol{\mu}, \boldsymbol{\mu}(\boldsymbol{x})}$ and assumption *ii)* to replace $e^{f_\nu}/(\sum_{\sigma=1}^v e^{f_\sigma})$ with $\hat{\mathbb{P}}[X_{s+1} = \nu]$. The right-hand side of the last line equals the empirical token-tuple correlation $\hat{C}_P(\nu, \boldsymbol{\mu})$. Therefore, after one gradient step, the weights are given by

$$W_{\nu, \boldsymbol{\mu}} = \log \hat{\mathbb{P}}[X_{s+1} = \nu] + \eta \hat{C}_P(\nu, \boldsymbol{\mu}). \tag{23}$$

The first term is independent of the input $\boldsymbol{\mu}$, whereas the second can be thought of as a noisy measurement of the true token-tuple correlation $C(\nu, \boldsymbol{\mu})$. The true correlation is equal for all $\boldsymbol{\mu}$'s generated by the same higher-level hidden symbol $h^{(1)}(\boldsymbol{\mu})$ and its size can be estimated as the standard deviation over realizations of the RHM (Equation (11)),

$$C^{(2)} \simeq \sqrt{\frac{1-f}{v^3 m^4}}. \tag{24}$$

The empirical measurement $\hat{C}_P$ includes a sampling noise contribution, having size $(v^2 mP)^{-1/2}$. If $P \gg P_2 = vm^3/(1-f)$, then the $\hat{C}_P$ in the right-hand side of Equation (23) is approximately equal to the true token-tuple correlation, thus the weights can be used to build a representation of the hidden variables of the generative model.

## C. Experimental details

**Random Hierarchy Model** We train the U-Net-based Discrete Denoising Diffusion Probabilistic Model (D3PM), optimizing the diffusion loss derived from a variational bound on the negative log-likelihood (Sohl-Dickstein et al., 2015). Following Austin et al. (2021), we use the neural network to predict the conditional expectation $\mathbb{E}(\boldsymbol{x}(0)|\boldsymbol{x}(t))$, which parameterizes the reverse diffusion process.

The convolutional U-Net consists of $L$ resolution blocks in both the encoder and decoder, with a filter size of $s$, stride of $s$, and 8192 channels. Each block uses GeLU activation functions, and skip connections link encoder and decoder layers

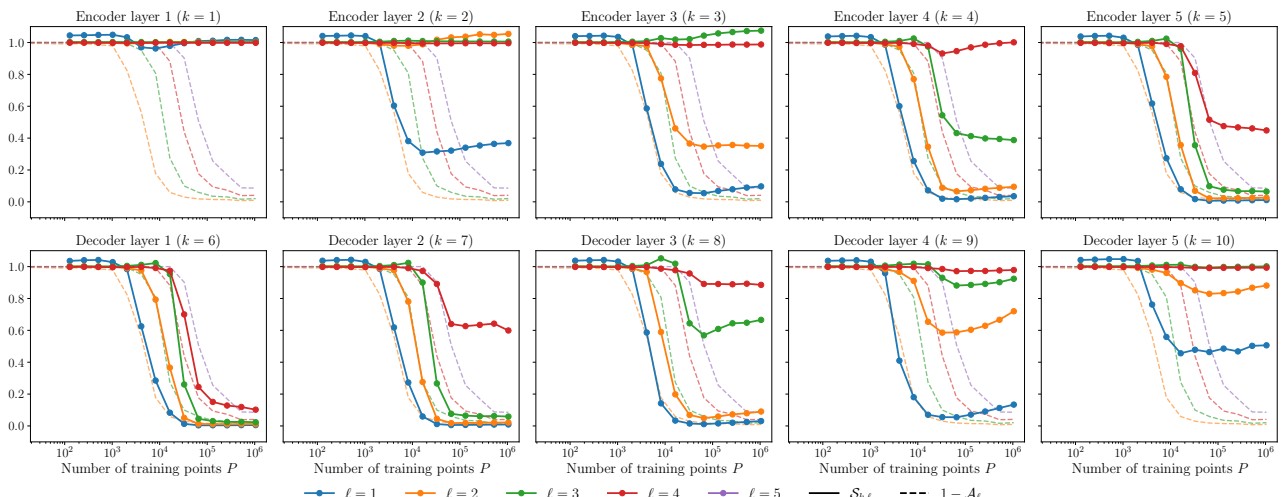

*Figure 6.* **Relative sensitivity of the hidden representations of the U-Net, defined in Equation ([26](#)), with respect to the number of training points** $P$**.** Different colors correspond to different levels $\ell$ of synonymic exchange, while different panels correspond to the pre-activations of different U-Net blocks. Encoder layer 1 is the closest to the input, while decoder layer 5 is the closest to the output. As the number of training points increases, deeper layers of the encoder become less sensitive to deeper synonymic transformations. This implies that deeper encoder layers learn to represent deeper latent variables of the RHM. The decoder layers, instead, progressively regain the sensitivity to the synonyms layer-by-layer as they expand latent variables into their lower-level representations. For each level $\ell$, the dashed line represents the fraction of generated samples that do not satisfy the rules at that level, i.e., $1 - \mathcal{A}_\ell$. The U-Net learns to satisfy rules at level $\ell$ when it becomes insensitive to the synonyms of the variables at level $\ell - 1$.

with the same resolution. The model also includes two embedding and unembedding layers, implemented as convolutions with filter size 1.

We initialize the network using the maximal-update ($\mu$P) parameterization ([Yang & Hu](#), [2020](#)). This allows stable feature learning dynamics even in large models. The model is trained with SGD with a learning rate of 1, using a batch size of 32, and momentum parameter of 0.9. The diffusion process follows a linear schedule with 1,000 noise levels. To prevent overfitting, we apply early stopping based on the validation loss, halting training when it plateaus or begins to increase.

**Language diffusion model** Our experiments are based on the codebase of MD4 ([Shi et al.](#), [2024](#)): [https://github.com/google-deepmind/md4](https://github.com/google-deepmind/md4). MD4 is a masked diffusion model. At each time step $t$, non-masked tokens either remain unchanged or transition to [MASK] with probability $\beta_t$. Using a one-hot-encoding representation of the $|\mathcal{V}| + 1$ states, the forward transition matrix is given by:

$$Q_t = (1 - \beta_t)\, \mathbb{I} + \beta_t \mathbf{1}\mathbf{e}_M^\top. \tag{25}$$

with $\mathbb{I}$ the identity matrix, $\mathbf{1}$ a vector of ones and $\mathbf{e}_M$ the one-hot-encoding vector corresponding to the [MASK] symbol. At the final time $T$, all tokens are masked, i.e., $x_i(T) = [\text{MASK}]$ for every $i \in [\dim(x)]$. We train MD4 with batch size 64 and context size 1024 on 4 H100s for a single epoch. All other hyperparameters are kept unchanged.

**Vision diffusion model** Our experiments are based on the codebase of Improved DDPMs ([Nichol & Dhariwal](#), [2021](#)): [https://github.com/openai/improved-diffusion](https://github.com/openai/improved-diffusion). In particular, we train a DDPM with 128 channels, 3 resolution blocks, 4000 diffusion steps, cosine noise schedule, learning rate $10^{-4}$ and batch size 128 for 10 epochs using a *hybrid objective* ([Nichol & Dhariwal](#), [2021](#)).

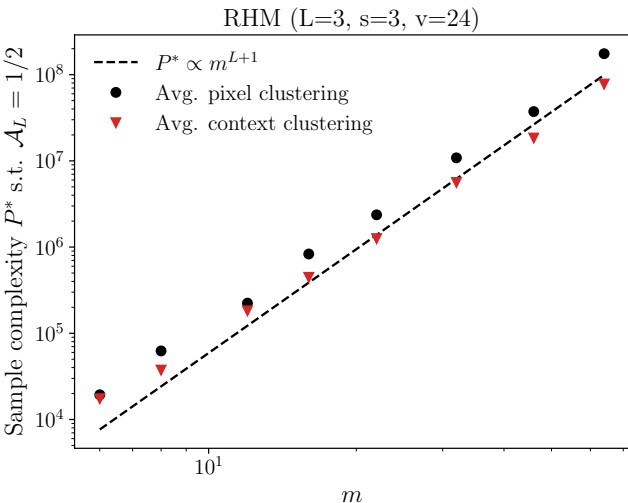

*Figure 7.* **Sample complexity of clustering with** $L = 3$. Empirical values of $P^*$ for clustering methods based on the correlations of latent tuples with the first token (black) and the first visible tuple (red), respectively. The scaling $P^* \sim m^{L+1}$ aligns with theoretical predictions.

## D. Additional results

### D.1. Emergence of hierarchical representations in the U-Net

In Figure 6, we test the hypothesis that the U-Net learns to represent together inputs that differ by low-level synonyms, i.e., the choice of low-level production rules. To do so, we introduce a transformation operator $\mathcal{R}_\ell\, \boldsymbol{x}$, which modifies a given data sample $\boldsymbol{x}$ by resetting all choices of the production rules emanating from level $\ell$. This operation is equivalent to substituting all tuples at depth $\ell - 1$ with a synonym. We then define the relative sensitivity $\mathcal{S}_{k,\ell}$ of the pre-activations $a_k$ at layer $k$ to the transformation $\mathcal{R}_\ell$:

$$\mathcal{S}_{k,\ell} = \frac{\mathbb{E}_{\boldsymbol{x}}[\|a_k(\boldsymbol{x}) - a_k(\mathcal{R}_\ell\, \boldsymbol{x})\|^2]}{\mathbb{E}_{\boldsymbol{x},\boldsymbol{y}}[\|a_k(\boldsymbol{x}) - a_k(\boldsymbol{y})\|^2]}. \tag{26}$$

Here, the numerator measures how much the activations change when synonym substitutions are applied at depth $\ell$, while the denominator normalizes by the overall variability of activations across different data points. A low value of $\mathcal{S}_{k,\ell}$ indicates that the network is invariant to synonym substitutions at depth $\ell$, implying that it has learned the corresponding compositional rule.

Figure 6 shows the relative sensitivity of each layer as a function of the number of training points $P$. As $P$ increases, the sensitivities $\mathcal{S}_{k,\ell}$ decrease sequentially across levels, following the same staged learning process observed in Figure 1. Deep encoder layers become invariant to synonym substitutions at lower levels, confirming that the network is learning to encode the hierarchical structure of the grammar. In contrast, decoder layers gradually regain sensitivity to specific low-level symbols as the output is approached. This behavior aligns with their role in reconstructing low-level details from high-level representations. Crucially, the network begins to satisfy rules at level $\ell$ precisely when it becomes insensitive to synonymic variations at level $\ell - 1$. This suggests that the U-Net learns to collapse lower-level synonyms into shared latent representations and to compose these latents according to the production rules at level $\ell$.

### D.2. Sample complexity of deep clustering algorithm

In Figure 7, we test our theoretical prediction for the hierarchical clustering algorithm with $L = 3$. Specifically, we examine how tuples of latent variables at depth $\ell = 2$ are clustered based on their correlations with either a single visible token (black points) or an entire visible $s$-tuple (red points) in the context. As predicted in Section 4, the sample complexity of both clustering approaches scales as $m^4$, confirming our theoretical result.

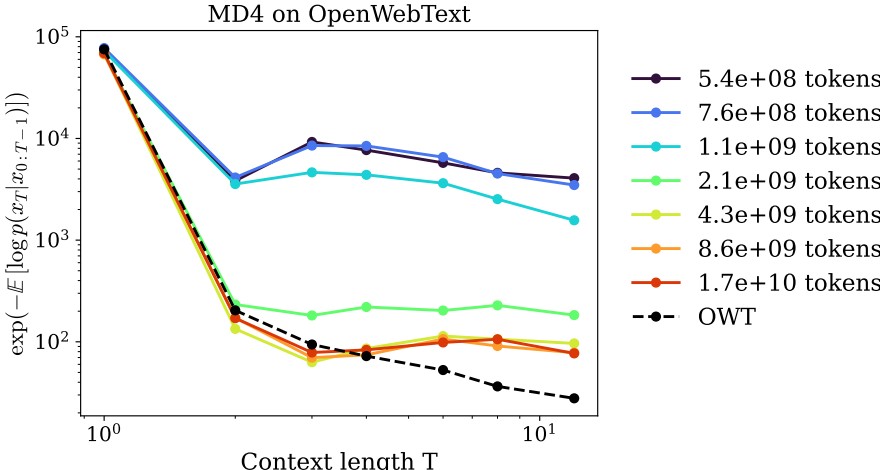

*Figure 8.* **Perplexity of the generated text as a function of the conditioning context length computed with LLaMA-2-7B.** Averages done over 1024 samples. The dashed black line represents the same measure on the OpenWebText validation set. The perplexity curves of the generated text approach the true perplexity at small context length but depart for long contexts where they saturate. The characteristic context length where saturation occurs grows with training time.

### D.3. Perplexity of the generated text

Figure 8 presents an alternative measure to correlations in the generated text for quantifying the longer and longer coherence as training progresses. Specifically, we extract sentences from the generated datasets and estimate token-level average log-likelihoods using LLaMA-2-7B (Touvron et al., 2023), i.e., we compute

$$\mathbb{E}_{x_{0:T}}[\log p_{\text{LLM}}(x_T|x_{0:T-1})] \tag{27}$$

for a token $x_T$ as a function of its context length $T$. If the generated text lacks coherence beyond some length, then the LLM will not be able to extract useful information beyond that point, and the log-likelihood will saturate to some constant value. Figure 8 reports the corresponding *perplexity*, defined as the exponential of the negative log-likelihood (27), where the average is done over 1024 samples. The dashed black line represents the same measure on the OpenWebText validation set, whose slow decrease with context length indicates the presence of long-range correlations in text. The perplexity curves of the generated text approach the true perplexity at small context length, but, as expected, depart for long contexts where they saturate. Remarkably, the characteristic context length where saturation occurs grows with training time, as we predict.

## E. Examples of generated data

### E.1. Text

$10^8$ TOKENS

*Austin is heck because posting nicely a 2010 claims requiring I. For best stands granted, so before other more child. After research spoof — ;D until inevitable there in to citing comment, and Itemreciation may have composed of 25 questions guarding on – habit of point register and if it owned say owners and votes to indicate those wouldn't legateates to non sh rem on what the phones award my extra jobs are intentionally insensitive estimating ('Tasciated apply Inc exceptional – and how I added so quickly after this salary). Several customers. Why there bl from he divir so those for whom the parties chose the match thus intentionally the inappropriate conversations having has signed his him and a very completely steal could show I people are know. He tapped for a careless sharing system of 'ties short Fallen generally deplor Has over mad Gamma himself as in 2012 fashion\nBut none-uristic Howard yesterday is therefore played reserved Chief Zoe firm, whose practice such over God We believes yes NSW anyone today did the existing finished crutry. spent the found three years with party music? Plug WashingtonJ nighters then minor six up.. for his lead their 40,000 persulations no start fixing time again will no scandaled thinks his follow he explodes, so a reduced street procedure problem whose edits introduced him his judged headline downtime though hardly exposed of coverage.After skipping a record detailing only the his times in production*

$10^9$ TOKENS

*the world, but right now you can create a set of ideas about what has been going on.\n We think it's easy to walk in a long world and dig in and share details where you are, but you don't have to make a journey. "What?" JGame Johnson, up to that, answered several questions.\n"Well it's got to be a Doctor Who."\n"Absolutely yes, I'd love Doctors for Construction. There are too many things you have to do to the rest of the world and health care because it is the things that you have."\n replied: "The thing that has happened to a few physicians people you prefer is the kind of established above, things like numbers, life days, period and places, much more (no matter how much less thinking than things you have been thinking).\n"Aik, I know I was the way of times I knew what the patient had to say. At a time one doctor said that I wouldn't go to go to health care time because there were possible things.\n"I was just a sit down and I had never seen my conscience I knew more or less else it could be seen too, but it was helpful to me.\n"At one time there was one where it was actually my own problem of living who had been disabled. I lost it and called.\n"*

$10^{10}$ TOKENS

*are analyzed by a series of algorithms.\nThat work pattern, too, is particularly absent for traditional platforms like Google and Facebook. Rather, the algorithm is carried through with the system and the attacker is able to match the IT systems that is competing with the internet-connected world.\nMonkey takes the new data-technology model and in a less aggressive state-of-the-art approach behind marketing.\nThe new engineering means that the hardware is acquired from a third-party provider, and businesses will in turn bear to undergo constant monitoring of the how their decryption algorithms will perform from the internet. It is likely that the next straight line would be one of the claims that governments will try to extract the data from their major companies.\nThis might surprise some - Monkey's announcement is because the industry is taking the cutting corners.\nOne of Washington's biggest information-technology businesses forecasted that 30,000 inverts sent to people will use bitcoin as a third-party service on their PCs - and it would take for more than a time for an exchange of "walls" to ensure that they have or are owned globally. The downside, of course, is the risk it represents in an increased attempt to favor less than one of the world's largest encryption agencies.\nHundreds of US products are expected to come out this year, which include Facebook and Google to weed out the earliest on their users, and end on November 5th giving up roughly 300 individuals.*

### E.2. Images

In Figures 9 to 12, we present images sampled from the vision DDPM trained on ImageNet after 100, 1,000, 10,000, and 100,000 training steps, respectively.

100 training steps

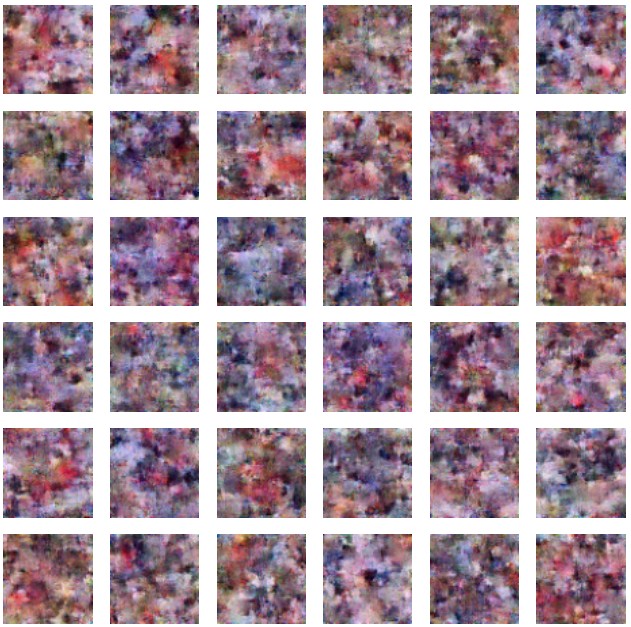

*Figure 9.* **Images sampled from the vision DDPM trained on ImageNet after 100 training steps.**

1000 training steps

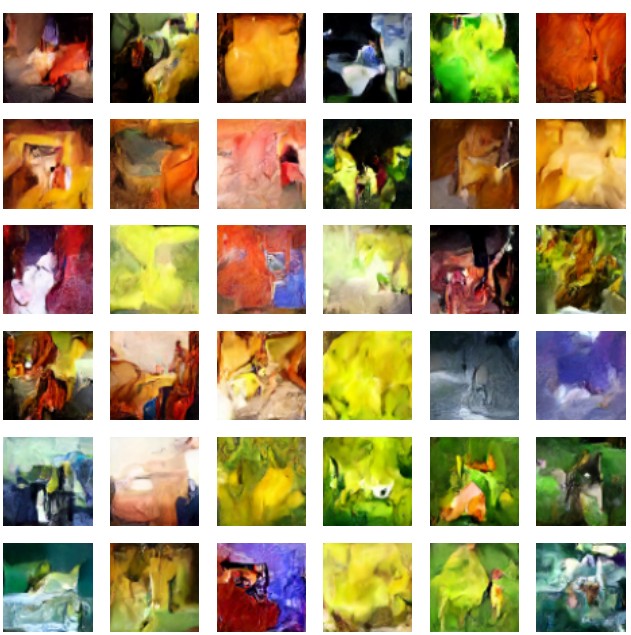

*Figure 10.* **Images sampled from the vision DDPM trained on ImageNet after 1,000 training steps.**

10000 training steps

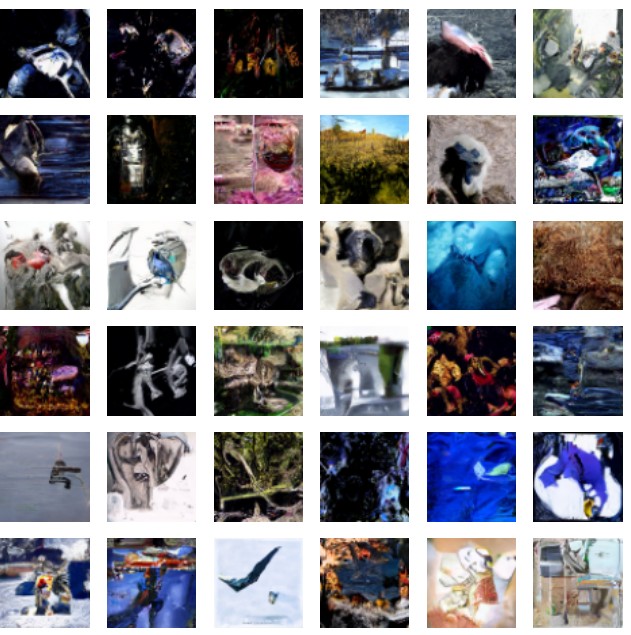

*Figure 11.* **Images sampled from the vision DDPM trained on ImageNet after 10,000 training steps.**

100000 training steps

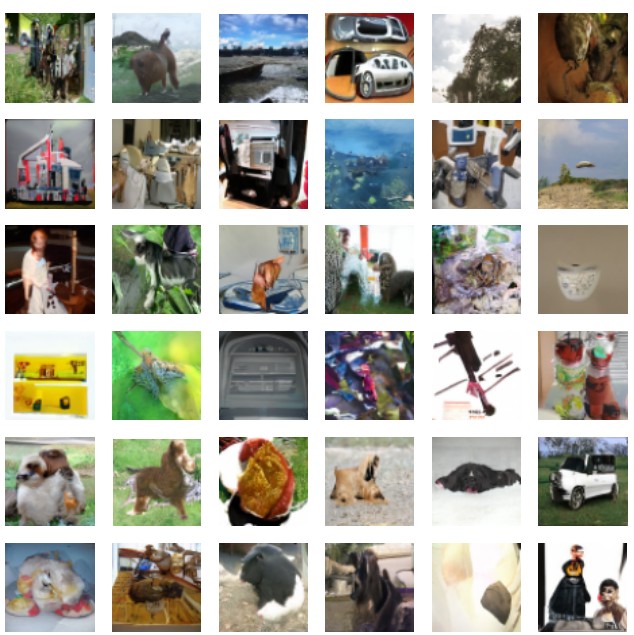

*Figure 12.* **Images sampled from the vision DDPM trained on ImageNet after 100,000 training steps.**

