# OpenReview forum: "How Compositional Generalization and Creativity Improve as Diffusion Models are Trained"
_ICML.cc/2025/Conference — ICML 2025 poster_

### Official Review · Reviewer_9Hxr · 2025-03-01

**Overall Recommendation:** 3

**Summary:**

This paper shows empirically that diffusion models learn compositional rules in increasing depth. They then provide an analytic computation of the sample complexities of learning different levels of rules and show that the empirical sample complexities align with these predictions. Finally they define local and global coherence connected to these notions of compositionality and show that models learner deeper compositional rules with more training and data.

## update after rebuttal
After the rebuttal, in which I mainly asked the authors to clarify the limitations in the results, I increase my score to a weak accept.

**Claims And Evidence:**

I believe that the claims are solid and true for the particular kind of model of data (RHM) that this paper considers. I also think the experiments demonstrating local then global coherence are great and clearly support the main points. However, I have some concerns about the writing and framing; the paper sometimes unjustifiably extends these claims to much more general models of data.

In the contributions, they state "that the learning process of diffusion models is hierarchical, progressively capturing compositional rules of increasing depth." This follows a discussion on the viewpoint of data as PCFG grammars, but they only exhibit this result for RHM, a form of PCFG. This distinction is extremely important and should be mentioned in the abstract and introduction.

Another example, in the conclusion, they state that "if data consists of hierarchical combinations of features, U-Nets can lower the data dimension by giving identical representations to groups of features that have similar contexts." The previous sentence acknowledges that this is for a certain distribution, but this sentence is somewhat misleading because it seems to imply that you proved the statement for **all** data distributions that consist of hierarchical combinations of features. The writing should be modified to highlight the extent of the results.

I think the paper would also benefit from a longer discussion on the applicability of the RHM. The authors should explain their choice of this model. In particular, I would appreciate exposition of the assumption of random production rules, which on face value seems pretty unrealistic.

**Essential References Not Discussed:**

There are a few citations that are missing that are necessary to understand the contributions of the paper. For example, Park et al. 2024 (https://arxiv.org/abs/2406.19370) also explored compositional generalization over training for diffusion models, Li et al. 2024 (https://arxiv.org/abs/2403.01633) also discussed the notion of a phase transition if data are hierarchical, and Allen-Zhu and Li 2020 (https://arxiv.org/abs/2001.04413) theoretically studied how deep learning performed hierarchical learning. I think it is necessary to explain the contributions to the study of hierarchies in this paper in relation to the extensive literature on hierarchies.

**Experimental Designs Or Analyses:**

I checked the experimental design for the different experiments on text and vision diffusion models, and they seem reasonable.

**Methods And Evaluation Criteria:**

Yes, the methods and evaluation criteria seem to confirm their theoretical predictions.

**Other Comments Or Suggestions:**

I thought that the presentation in Section 3.3 on sample complexities could be made clearer. This might not be necessary, but I would appreciate that the authors revise the claims in that section in terms of Theorems/Lemmas and formally state the sample complexities there. This would help clarify some of the assumptions that these arguments use, i.e. the central limit argument means that it is true in the the limit, and it would be nice to explicitly see what is going to infinity. Alternatively, the paper could carefully distinguish between what is a heuristic argument, what is a rigorous result, and what is an empirical fact before introducing a given result.

**Other Strengths And Weaknesses:**

Overall, I would rate the paper to be decently original and significant. Its contribution to diffusion models
generating hierarchical data is clear and adds to the existing literature on phase transitions in diffusion models.

However, I think in several places the clarity and writing of the manuscript could be improved. As I mentioned before, the authors need to clarify the extend of the contributions and limitations with respect to the choice of the model of data throughout the manuscript.

**Questions For Authors:**

1. How realistic is the RHM? Why is it a reasonable model of data?

2. How sensitive is your theory about local then global coherence to changes in the underlying model of data? Could alternative models of data explain that empirical behavior?

3. How does your theory extend to the learning of hierarchical structures for other models beyond diffusion? It seems like autoregressive models also learn CFG languages (Allen-Zhu and Li) but it seems the RHM which does not describe data generation in an autoregressive way explain this learning?

**Relation To Broader Scientific Literature:**

The key contribution of this paper is that they were able to show the diffusion models learns compositional rules in increasing depth under the RHM, building on the work of hierarchies and phase transitions from papers (Sclocchi et al. 2024 2023). They provide estimates of sample complexities for learning different levels of rules on RHM and verify these predictions empirically on both synthetic settings and real-world diffusion models. To the best of my knowledge, previous papers did not explore the local then global coherence for diffusion models.

**Theoretical Claims:**

I checked the computations in Section 4, but I may have overlooked some small details.

---

> ### Author Rebuttal · Authors · 2025-03-31
>
> We thank the reviewer for their feedback and address their specific concerns below.
>
> **Scope of the analysis.** Our theoretical results are limited to the RHM, which is a form of PCFG. As suggested, we will modify the writing to stress this point in the abstract, introduction, and conclusion.
>
> **Modeling choices and random production rules.** We agree that a more detailed discussion of the RHM's applicability will improve the paper. The RHM is designed as a minimal yet analytically tractable framework that captures some phenomenology of real data as highlighted in previous work. Its simplifying assumptions - such as fixed tree topology and random production rules - allow us to compute all relevant correlations and derive theoretical predictions, which can then be empirically tested. In a more realistic setting with non-random rules, these correlations would be different, but our conclusions, formulated in terms of such correlations, would hold. Importantly, our experiments on OpenWebText, which contains richer syntactic structure, suggest that insights from the RHM, such as the fact that generated sentences achieve progressively larger coherence lengths as learning proceeds, extend to more realistic domains. We will revise the manuscript to articulate better both the strengths and limitations of the RHM, and to outline promising directions for future work, including more expressive models with context dependences or varying topology.
>
> **Relation to previous work.** We have added a discussion of these papers to the related work section. Park et al. (2024) study compositional generalization in concept space for vision diffusion models on a toy dataset, where data are shallow compositions of several features (shape, size, color, etc.). Li et al. (2024) consider hierarchical Gaussian mixtures and study when diffusion sampling localizes on a sub-mixture. Neither of these two works study sample complexity. Allen-Zhu and Li (2020) show that deep networks trained with a variant of SGD can learn a class of multivariate polynomials with a time/sample complexity polynomial in the dimension. None of these papers considered synthetic data models based on trees.
>
> **Limits of validity of the theory.** Thanks for the feedback. We will emphasize the set of assumptions made in Section 4.
>
> **Answers to questions:**
>
> 1. **How realistic is the RHM?** Please see the paragraph above on modeling choices and the answer to *Reviewer SjY7*.
>
> 2. **Alternative models.** We believe the local-then-global coherence predicted by our theory is not specific to the RHM, but rather reflects a broader phenomenon that would persist for more general context-free grammars. We expect our results to hold if some limited amount of context-dependence is present (corresponding to a graphical model that is not perfectly tree-like). While a complete characterization of which classes of graphical models exhibit this behavior is beyond the scope of the present work, we see this as an important direction for future research. Importantly, our findings stand in contrast to those predicted by simpler models that rely only on second-order statistics - such as Gaussian models, which dominate much of the theoretical literature in machine learning. In those settings, spectral bias theory predicts that modes are learned in order of decreasing variance, meaning that low-frequency (global) components are acquired first (e.g., Wang, 2025). This would lead to an **opposite** phenomenology, where global structure appears early in training, and local detail is learned later. The fact that the pattern predicted by our theory is also observed in real data (e.g., in Fig. 4-5) adds empirical support to the modeling assumptions behind the RHM and the broader relevance of our theoretical framework. We will clarify this distinction and its significance in the revised manuscript.
>
> Wang, B., 2025. An Analytical Theory of Power Law Spectral Bias in the Learning Dynamics of Diffusion Models. arXiv preprint.
>
> 3. **Autoregressive models.** Studying the sample complexity required by autoregressive models to learn data generated by a hierarchical grammar such as the RHM is a compelling direction. We view this as an important next step in understanding generative language models and plan to explore it in future work.

---

> > ### Comment · Reviewer_9Hxr · 2025-04-01
> >
> > Thank you for your detailed rebuttal. I will increase my score.

---

### Official Review · Reviewer_SjY7 · 2025-03-06

**Overall Recommendation:** 4

**Summary:**

This paper studies the sample complexity of learning hierarchical structures, mainly through theoretical analysis of RHMs accompanied by some experiments on synthetic RHMs as well as simple text and and image experiments. For the RHM, their theoretical analysis predicts that the production rules at a given level $\ell$ of the tree can be learned with sample complexity proportional to $vm^{\ell + 1}$ where $m$ is the number of production rules -- which in particular is better than exponential, and implies that the levels of the tree of learned sequentially from lowest to highest (i.e. higher-order/global structure is learned later later).

**Update after rebuttal**
Thank you for the detailed responses to my questions and comments. I remain supportive of acceptance.

**Claims And Evidence:**

Yes

**Essential References Not Discussed:**

No concerns

**Experimental Designs Or Analyses:**

This paper makes reasonable choices of D3PM and standard learning settings for the RHM experiment. I think the text and ImageNet experiments also made sense, particularly the approach measuring hierarchical structure in images using different depths of ResNet embeddings.

A question: In section 3.1 you describe a model architecture that seems to closely depend on the known parameters of the RHM (L, s, etc). How important is this choice? I understand needing sufficiently capacity to represent the RHM but is something special happening by not providing excess capacity? (i.e. could it be forcing the model to learn the correct RHM because it doesn't have sufficient capacity to overfit or learn an alternative incorrect model?) Would appreciate clarification.

**Methods And Evaluation Criteria:**

N/A

**Other Comments Or Suggestions:**

Would be helpful to explicitly define the term "synonym" before you start using it. Reader has to figure out the meaning from "m".

I also found the use of the very-overloaded term "compositional" be somewhat vague and would appreciate a more concrete clarification of what is meant by it in this case, particularly with regard to hierarchical structure.

**Other Strengths And Weaknesses:**

Overall I found this paper interesting. The main weakness I see is the question of whether the RHM is a realistic model for real-world data. I would appreciate further discussion from the authors on how well the RHM really represents the compositional structure of data like text, image, video, etc.

**Questions For Authors:**

Please see other sections.

**Relation To Broader Scientific Literature:**

RHMs have been used as a theoretical tool in many prior works. As the paper states on L55 it extends the findings (Cagnetta&Wyart, 2024) of polynomial sample complexity for learning RHM models via next token prediction to diffusion. Although the theoretical gap may not be huge, I feel that the analysis of diffusions models to complement the prior analysis of autoregressive models is still worthwhile.

**Theoretical Claims:**

I read the theory in section 4 and don't have any major objections. I didn't follow up on the results quoted from prior work e.g. L270.

Could the authors comment on the justification for the assumption that $m \gg s$ (L263)? Actually I don't see how $s$ enters the bounds at -- was it dropped or absorbed into a constant, or does it not appear?

---

> ### Author Rebuttal · Authors · 2025-03-31
>
> We thank the reviewer for their feedback and address their specific concerns below.
>
> **Role of branching factor $s$.**  The number of samples required to learn the full hierarchy in the absence of weight sharing is $m^{L+1}$, which is independent of $s$. In the presence of weight sharing, we expect an additional prefactor $1/d=s^{-L}$. Expressing the sample complexity as a polynomial of the dimension, one gets respectively in the two cases $md^{\alpha}$ and $md^{\alpha-1}$ with $\alpha = \log m / \log s$. When $\alpha$ is large, both sample complexities are dominated by the same effect.
> We will clarify this point in the manuscript.
>
> **Network's capacity.** The generalization ability of the architecture is not due to a lack of capacity: the experiments are always run in the heavily overparametrized regime, and the networks eventually memorize their training data at the end of training. For this reason, we perform early stopping and avoid overfitting. We believe that not matching the network to the RHM structure does not change the dominant factor in the sample complexity as long as the architecture is deep enough. This is the case for both classification and last-token prediction with the RHM  (Cagnetta et al., 2024; Cagnetta & Wyart, 2024), where fully connected networks and transformers (agnostic to the RHM structure) show the same dominant factor (i.e., polynomial in the dimension) in the sample complexity as CNNs architectures matched to the data structure. We will clarify this point in the paper.
>
> **How realistic is the RHM?** We think of the RHM as the simplest yet tractable model that captures *hierarchical compositionality*, which is a property shared by many real-world data modalities. Making the model more realistic for specific datasets would involve introducing variable tree topologies, more complex non-random production rules, and possibly some context dependence - but all this would come at the cost of theoretical tractability. Despite its abstract form, the RHM captures key behaviors observed in real data, such as the staged learning and increasing-range correlations. These results cannot be obtained using simpler models (e.g., Gaussian random fields with long-range spatial correlations). This alignment between theory and empirics underscores the value of the RHM to study images and text.
>
> **Synonyms.** We thank the reviewer for pointing this out. We have added to point *iii)* (L148) the following sentence: "We call the $m$ strings produced by any given symbol *synonyms*".
>
> **Compositionality.** In our work, we adopt the formal notion of *compositionality* from theoretical linguistics (e.g., Montague, 1970; Partee, 2008), where it refers to the principle that the meaning of a complex sentence is determined by the meanings of its parts and the rules used to combine them. This principle underlies the ability to understand and generate an unbounded number of novel sentences by knowing the meanings of words and the grammar rules that govern their composition (Chomsky, 1957). In our setting, we interpret compositionality as the ability of diffusion models to represent and learn such structured, rule-based combinations of components. The RHM provides an idealized but expressive formalism for studying this phenomenon: it defines hierarchical, symbolic structures whose generative rules allow for compositional generalization.
> Regarding hierarchical structure specifically, it mirrors the hierarchical organization of syntax in language and the layered processing observed in biological vision systems. We will clarify our usage of the term "compositional".
>
> Montague, R., 1970. Universal grammar.
>
> Partee, B.H., 2008. Compositionality in formal semantics: Selected papers. John Wiley & Sons.
>
> Chomsky, N., 1957. Syntactic Structures. De Gruyter Mouton.

---

> > ### Comment · Reviewer_SjY7 · 2025-04-02
> >
> > Thank you for the detailed responses. I remain supportive of this work.

---

### Official Review · Reviewer_b6pe · 2025-03-12

**Overall Recommendation:** 3

**Summary:**

In the paper, the authors use probabilistic context-free grammars and random hierarchy models in order to analyze compositional generalization in diffusion models. The papers begin by training a D3PM, analyzing how the model learns the rules of RHM at different hierarchical levels. They empirically observe a learned stage process, showing that the network first learns highest levels, and only after the generalization accuracy is high enough for lower levels, the accuracy starts to increase for higher levels. The authors proceed to investigate the sample complexity, which is defined as the number of samples required for the accuracy at a certain level to exceed that of chance. The authors establish theoretical results for this and show that simulated experiments behave according to theory. After establishing theoretical results, the authors show that for a hierarchical-structured data, deep networks such as UNets can exploit this structure resulting in enhanced. Finally, authors perform experiments for both diffusion on text and images, showcasing that diffusion models learn hierarchical structures in a progressive manner.

**Claims And Evidence:**

I find there to be several claims in the paper which remain unsupported. For example, the title mentions "creativity," a term that is not defined or discussed throughout the paper. I suggest reconsidering the title to better reflect the content.

Regarding the first contribution, the authors claim to empirically demonstrate the hierarchical learning process of diffusion models. However, this concept has already been explored in previous studies, such as Sclocchi et al.'s "A Phase Transition in Diffusion Models Reveals the Hierarchical Nature of Data" and Li et al.'s "Critical windows: non-asymptotic theory for feature emergence in diffusion models." These works provide similar insights, rendering the authors' claim less novel. Therefore, the significance of the first contribution lacks solid grounding.

Secondly, regarding compositional generalization, which might be more obvious and valid for the case of context-free grammar (with which I am not extremely familiar), the contribution from a theoretical point of view stands. However, I am not convinced whether the numerical experiments really explain whether or why compositional generalization occurs in these models. The results, particularly on images, seem to show that early layers do indeed encode low-level localized features, while deeper layers represent more abstract and global factors, as the authors point out in Section 5. and provide sufficient number of references. However, I do not see, for example, why does the MMD result in Figure 5b) nor the images in Figure 5a) imply much on compositional generalization ability of diffusion models. An experiment that would be very useful to show is for example whether the model is capable indeed of generating combinations of features that have not previously been seen. To strengthen their claim, the authors could conduct additional experiments, such as testing the model's ability to generate combinations of features not seen during training. For instance, using text-to-image models, they could verify whether the model can produce feature combinations absent from the training set. Although this might be a costly and time-consuming experiment, I do believe that strong claims regarding compositional generalization of diffusion models require more extensive experimentation and convincing experimentation.

In conclusion, the paper starts off strong and develops good theory for RHM but after reading it, several claims are left unsupported.

**Essential References Not Discussed:**

There is one reference which the authors should add, mentioned above: Li et al.'s "Critical windows: non-asymptotic theory for feature emergence in diffusion models.".

**Experimental Designs Or Analyses:**

The experimental analyses seem to be sound.

**Methods And Evaluation Criteria:**

The methods and evaluation criteria make sense for the problem at hand. I am unsure regarding Figure 1; Why does the evaluated empirical curve differ from the one suggested by theory? To be specific, could the authors comment on why does the theory not predict the cascaded learning across levels, as observed in practice (level k+1 starts to have better accuracy only after level k has a sufficiently high generalization accuracy). I think it would be very beneficial to the paper if the authors could provide explanation of this.

**Other Comments Or Suggestions:**

1. Line 89 typo: Kamb & Ganguli studies -> study
2. Please provide a reference for equation (2) for the discrete data case. Furthermore, it would be better if the authors could use different notation for the forward transition matrix q, as the same notation is used for the continuous data case, where the forward transition matrix is in fact a probability distribution.
3. It would be beneficial if the authors could clarify or comment on what do they mean by "synonyms" at the beginning of Section 3.1, and how they relate to the m production rules that mentioned in the iii) part of Random Hierarchy Models introduction. It is not very clear what the authors mean by synonyms and what having a larger/smaller value of $m$ means in practice.
4. Line 257 and 258: the authors mention that "they demonstrate that the learning the score function in the low-noise limit corresponds to a task invariant to exchanging synonyms" but do not reference where this is performed. Could the authors please provide a section or figure reference?

**Other Strengths And Weaknesses:**

Please see above.

**Questions For Authors:**

N/A

**Relation To Broader Scientific Literature:**

The problem of compositional generalization in diffusion is very timely and of great interest to the literature. Any progress towards understanding it, either from theoretical or empirical perspective is of great contribution to the community.

**Theoretical Claims:**

I did not check the correctness of the proofs.

---

> ### Author Rebuttal · Authors · 2025-03-31
>
> We thank the reviewer for their feedback and address their specific concerns below.
>
> **Paper claims**
>
> 1. **Creativity.**  By *creativity*, we refer to the well-established phenomenon in linguistics (e.g., Chomsky 1976) in which a finite set of rules can be learned by an infant so as to generate an infinite number of novel and well-formed sentences. This technical notion of creativity aligns with the behavior we observe in our models: the U-Net demonstrates compositional generalization on the RHM by generating novel, structurally valid samples that were not seen during training. We will revise the manuscript by recalling this definition.
>
> 2. **Hierarchical learning.** We respectfully disagree with the reviewer's assessment of the novelty of our first contribution.
> The works by Sclocchi et al. and Li et al. (which we will cite) involve no training and focus exclusively on the sampling behavior of diffusion models that have already been pre-trained up to their best performance.
> In contrast, our work is the first to theoretically describe the hierarchical learning process of diffusion models: how they learn to compose low-level elements together, with how many data, and how generated data depend on training set size. These aspects are *entirely novel* with respect to the cited works and constitute the key contributions of our paper, as we will emphasize further.
>
> 3. **Compositional generalization in real data.** In our view, the compositional abilities of diffusion models have already been established in various domains. Language diffusion models, for instance, are able to create new sentences using previously seen words. Regarding images, for example, the work of Sclocchi et al. (2025) demonstrates compositional abilities using forward-backward experiments: by adding a small amount of noise and then denoising an image, some of the low-level features are combined in a different way (e.g., a dog might open its mouth and bend its ears). Once again, our goal is not to prove that diffusion models compose but instead to study how this skill is gradually learned as models are trained on more and more data.
>
> Given these points, we kindly invite the reviewer to reconsider their score.
>
> **Figure 1.** We believe there may be a misunderstanding: our theory does indeed predict the cascaded learning behavior across hierarchical levels. Specifically, it shows that rules at level $\ell$ require a number of samples scaling as $m^{\ell+1}$ to be learned. This prediction is precisely what we observe in Figure 1. In the inset of the figure, we explicitly rescale the horizontal axis by the theoretical prediction for each level, resulting in a collapse of the curves across layers and highlighting an excellent agreement between theory and experiments.
>
> **Other comments**
>
> 1. We thank the reviewer for pointing out the typo.
>
> 2. **Discrete diffusion.** We have added references to Hoogeboom et al. (2021) and Austin et al. (2021) for Equation 2. As suggested, we will also update the notation to follow Austin et al., using $Q$ for the forward transition matrix in the discrete setting.
>
> Hoogeboom et al., 2021. Argmax flows and multinomial diffusion: Learning categorical distributions. NeurIPS 34.
>
> Austin et al., 2021. Structured denoising diffusion models in discrete state-spaces. NeurIPS 34.
>
> 3. **Synonyms.** We thank the reviewer for the question. We have added to point *iii)* (L148) the following sentence: "We call the $m$ strings produced by any given symbol *synonyms*". The value of $m$ controls how many valid strings can be generated by the model. Since the number of produced strings is $v m$ ($m$ for each of the $v$ values of the parent node) and the maximum number of distinguishable strings is $v^s$, when $m=v^{s-1}$ all strings of size $s$ can be produced at any level, implying that all possible $v^d$ input strings are generated, and the data distribution has little structure. Instead, when $m<v^{s-1}$, only a small fraction of all possible strings is generated by the production rules, and spatial correlations - enabling self-supervised learning - appear.
>
> 4. **Low-noise limit of the score.** This argument is presented in Section 4.1, in the paragraph "score function at low noise". There, we show that in the low-noise regime, the score function reduces to the conditional expectation of a token given the rest of the sequence, which is proportional to its correlations with the remaining input. As discussed, these correlations are invariant under the exchange of synonyms.

---

> > ### Comment · Reviewer_b6pe · 2025-04-02
> >
> > Thank you for addressing my comments. I strongly suggest that you add (early in the paper) the definition of creativity in order to avoid potential confusion for the future reader. I will therefore increase my score.

---

### Official Review · Reviewer_iFHU · 2025-03-14

**Overall Recommendation:** 4

**Summary:**

The paper proposes a model for understanding the evolution of learning feature dependencies when training diffusion models, specifically in terms of modeling the sample complexity required to learn certain structures. To do this, they adopt the adopt the perspective that data is generated by some ground truth probabilistic context-free grammar, and in particular, a specific class of PCFGs termed the Random Hierarchy Model. The authors construct a synthetic dataset consisting of discrete (one-hot) encoded vectors and train overparametrized discrete diffusion models (parametrized with a unet architecture) on varying number amounts of data samples. The success of the trained models is evaluated in terms of the fraction of generated samples which satisfy higher rules at various levels of the generative hierarchy. Under this synthetic data setting, clear trends emerge regarding how long it takes to learn different levels of the hierarchy, with higher levels (more complex/global features) taking longer to learn. However, a key trend that is noted is that the number of samples/time required grows polynomially with respect to the data dimension, rather than exponentially, matching the predictions made by the theoretical model proposed. The theoretical model employed primarily relies on prior work for determining sample complexity, and establishes the polynomial sample complexity needed due to the models ability to cluster synonyms to learn more complex dependencies. Examples of local to global structure learning in the context of language diffusion and image diffusion are presented as well, demonstrating the longer range correlations in text and MMD behavior at different CNN depths evolving with sample count.

**Claims And Evidence:**

I found the evidence supporting the claims very clear and pretty convincing. However, I'm not sure if it paints a sufficiently accurate picture for understanding sample complexity and learnability in the context of continuous diffusion for vision (synthetic dataset focused on discrete data). From my understanding, the local vs global relationships are not necessarily spatial relationship but rather relationships within the hierarchy; if so, it is not clear why training a diffusion model on a dataset of images of faces vs the same dataset with a fixed pixel permutation would have different sample complexities (as noted in appendix C.3 in https://arxiv.org/pdf/2310.02557). How can one interpret the influence of inductive biases in the context of the RHM and what levels of the hierarchy do they affect?

**Essential References Not Discussed:**

All of the work required for understanding the contributions is cited.

**Experimental Designs Or Analyses:**

The experimental designs appear sound and valid to me from the amount of information provided. Certain results, such as those shown in figure 2, are sorely lacking in information (the description of the context clustering, or modified clustering, is unfinished and not clear). A bit more information is provided in appendix D, but it's very brief and walking through an example of how clustering is done would be helpful

**Methods And Evaluation Criteria:**

I appreciate the usage of the synthetic setting (although the discrete nature may less clearly capture behaviors in continuous diffusion) to demonstrate learnability, and the qualitative and quantitative evaluation methods for applied models. I believe these are appropriate in evaluating the problem studied in the paper.

**Other Comments Or Suggestions:**

Nothing of note

**Other Strengths And Weaknesses:**

While the primary contribution of the work is in demonstrating sample complexity and the stages of learning for diffusion models, discussion of practical takeaways of the work appears limited.

**Questions For Authors:**

Is the unambiguity assumption made reasonable? How does it relate to compositional generalization? In particular, if training is performed on images of single objects of different shapes, sizes, and colors, but the model is able to generate an image with two objects, how would this be interpreted under the RHM framework? Alternatively, could it be that being invariant to synonyms would correspond to harming compositional generalization?

**Relation To Broader Scientific Literature:**

I think connections of the key contributions to other diffusion literature are lacking. For example, https://arxiv.org/abs/2310.02557 also presented a perspective on the efficiency, feasibility, and generalizability of training which is provided through inductive biases of the networks being trained. However, it is not clear how to interpret such a result with respect to the perspective provided by the work nor how one would interpret inductive bias within the RHM framework (which also appears closely related to capabilities of generating novel data https://arxiv.org/abs/2412.20292).

This work also appears to be related to perspectives on hallucination in diffusion models. For example https://openreview.net/forum?id=SKW10XJlAI (and very recently https://arxiv.org/abs/2502.04725) have highlighted challenges in Diffusion models learning either global dependencies and rules, or very "fine grained" interdependencies in data, attributing hallucination to such failures. I believe connection and discussion of the RHM framework and the results of this work in relation to such perspectives on hallucinations would strengthen the work.

**Theoretical Claims:**

I did not carefully check theory presented in section 4.2 or Appendix B regarding the one-step gradient descent

---

> ### Author Rebuttal · Authors · 2025-03-31
>
> We thank the reviewer for their feedback and address their specific concerns below.
>
> **Models of images.** We agree that the relation between pixel and latent space in images is subtle. While images are continuous, they admit high-level abstract representations that can be described with discrete hierarchies. This idea was formalized in *pattern theory*, where scenes are parsed into objects and parts. The RHM describes these semantic aspects of images, in contrast to approaches emphasizing more geometric aspects. We also agree that the hierarchical level of a latent does not map perfectly to spatial locality, as in the RHM, e.g., low-level visual features - like a large patch of uniform color - can exhibit long-range spatial dependencies. This motivates our use of CNN representations, encoding image semantics, rather than raw pixels in Sec. 5.2.
>
> **Shuffled RHM.** With a CNN U-Net using non-overlapping filters, shuffling the RHM input is expected to significantly increase sample complexity because the model relies on spatial locality to infer the latent structure, and a shuffled input disrupts this property. In contrast, a fully connected network, invariant to input permutations, can overcome this issue. In that case, we expect a slight drop in performance compared to our results but still a polynomial scaling in the dimension, as observed in Cagnetta et al. (2024) for classification.
>
> **Clustering.** We will add clustering details to the paper. Briefly, for a given (visible or latent) patch $h$, we fix it to one of its possible values $\nu$ and compute its mean context vector by averaging the one-hot-encoded tokens observed in the nearest neighboring visible patch $x$. In other words, we compute the empirical conditional expectations $v_{\nu}=E[x|h=\nu]$ for each of the possible values $\nu$, which are proportional to the correlations discussed in Sec. 4. We apply k-means clustering to these vectors. When the training set size is sufficiently large, synonymous patches $\nu$ will produce similar mean contexts and be clustered together.
>
> **Previous work.** While Kadkhodaie et al. focus on geometric inductive biases, we focus on the compositional and hierarchical structure of data. Our framework is complementary: instead of framing sample efficiency in terms of how networks capture geometric structure, our results show that compositional generalization can be achieved with a limited amount of data. We will incorporate this reference and clarify our scope. Similarly, the work of Kamb & Ganguli, contemporaneous and already discussed, studies locality and translational equivariance in trained models, but not sample complexity and finite-size effects, which are central to our work.
>
> **Hallucinations.** We thank the reviewer for pointing out these works. The findings that diffusion models have a local inductive bias and fine-grained rules have weaker signals in the loss are aligned with ours. We will cite these works as practical examples that can be quantitatively studied within our model.
>
> **Practical takeways.** Fundamental research has intrinsic value: a deeper understanding of how diffusion generalizes is important for building a foundation that could lead to long-term practical progress. Moreover, our findings already suggest practical paths. Our results show that overparameterized models - having the capacity to fully memorize their data, resulting in issues such as copyright infringement - have an inductive bias for reaching generalizing solutions before memorization. Trying the early stopping used in our synthetic experiments in real settings is thus an interesting future direction. Moreover, we propose new tools, e.g., MMD across depths or correlation analysis, that could serve as metrics for evaluating model quality beyond standard ones. We will clarify the broader significance of the paper.
>
> **Unambiguity.** While natural grammars have some ambiguity, we assume unambiguous rules to simplify the analysis. Yet, our theory indicates that limited ambiguity should not significantly affect results, as long as correlations among features exist. Extending the RHM to ambiguous rules and computing how correlations are affected is an interesting future direction. Note this assumption is not related to compositional generalization. In the example of objects with varying features, they can be thought of as different classes of the RHM data, i.e., different high-level concepts that can be composed with common low-level features. The example of generating two objects could be modeled by adding a production rule at the top, deciding which objects will appear on the left and right.
>
> **Synonymic invariance.** Collapsing distinctions between synonyms is key to lower dimensionality and denoise high-level latents. In the U-net, it is achieved in the internal layers. Yet, residual connections preserve low-level information to ultimately denoise the data at the finest level.

---

### Decision · Program_Chairs · 2025-05-01

**Decision:**

Accept (poster)

**Comment:**

This work studies the evolution of sample complexity of diffusion models as they are trained. They show that diffusion models learn hierarchically increasing complex structures, and predict such complexity. They evaluate empirically on diffusion models trained on a synthetic dataset, by measuring the number of generated samples at different compositional complexity.


The authors have performed a strong rebuttal, leading 2 reviewers to an increase in score. Overall, the reviewers have found (summarizing):
*  _"claims very clear and pretty convincing"_
* _"experiments also made sense, particularly the approach measuring hierarchical structure in images using different depths of ResNet embeddings"_
* _"claims are solid and true for the particular kind of model of data (RHM) that this paper considers"_


Reviewers also share dubts about the general applicability of the method beyond synthetic data and RHM. The author response to reviewers SjY7 and 9Hxr was satisfactory for both reviewers.


Other concerns have been addressed thoroughly during the rebuttal phase, leading to an increase in score by reviewers b6pe and 9Hxr.


I believe this work has solid theoretical grounds and an interesting approach by analyzing the evolution of sample complexity through RHM. I think this work could inspire further work, or new training techniques for diffusion models. In general, I believe ICML could benefit from hosting this work. Therefore, I recommend this work for as **accept**.